# Compressed Sensing for Capability Localization in Large Language Models

**Anna Bair** [1]  **Yixuan Even Xu** [1]  **Mingjie Sun** [1]  **J. Zico Kolter** [1]

## Abstract

Large language models (LLMs) exhibit a wide range of capabilities, including mathematical reasoning, code generation, and linguistic behaviors. We show that Transformer architectures contain small subsets of attention heads that are necessary for certain capabilities. Zeroing out as few as five task-specific heads can degrade performance by up to $60\%$ on standard benchmarks measuring the capability of interest, while largely preserving performance on unrelated tasks. We introduce a compressed sensing-based method that exploits the sparsity of these heads to identify them via strategic knockouts and a small number of model evaluations. We validate these findings across Llama and Qwen models ranging from 1B to 14B parameters and a diverse set of capabilities including mathematical abilities and code generation, revealing a modular organization in which specialized capabilities are dependent on sparse, functionally distinct components. Overall, our results suggest that capability localization is a general organizational principle of Transformer language models, with implications for interpretability, model editing, and AI safety. Code is released at `https://github.com/locuslab/llm-components`.

## 1. Introduction

Understanding how large language models represent and execute diverse capabilities remains a central challenge in AI research. These capabilities, such as mathematical reasoning or code generation, represent higher-level skills that require coordinated computation across multiple model components, rather than simple fact retrieval. We investigate whether task-specific capabilities can be localized to specific components within Transformer architectures. Previ-

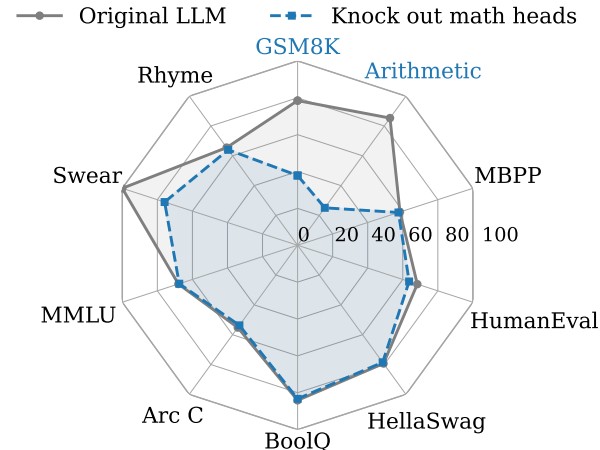

*Figure 1.* Knocking out the top five math heads identified by our compressed sensing method greatly reduces performance on math datasets (GSM8K and Arithmetic) while leaving other tasks relatively unaffected.

ous work has successfully isolated factual associations to particular neurons and layers (Maini et al., 2023; Meng et al., 2023), raising the question of whether the same principle of localization extends to complex behavioral capabilities. Our central finding is that many capabilities are indeed highly localized, but to sparse sets of attention heads rather than individual neurons.

Similarly to how factual knowledge can be traced to specific model components, we find that task-specific skills critically depend on small subsets of attention heads. We measure task-specificity by knocking out attention heads, or setting their output to zero, and evaluating the accuracy of the resulting model on the task of interest. A drop in accuracy indicates that the head was responsible for some amount of task-specific computation. Knocking out as few as five task-specific heads causes performance degradations of up to $60\%$ on standard benchmarks measuring the target task while leaving performance on unrelated tasks intact, as shown in Figure 1. This extreme localization suggests that Transformer models organize capabilities in a modular fashion, with different functional specializations relying on

[1]Carnegie Mellon University. Correspondence to: Anna Bair <abair@cmu.edu>.

*Proceedings of the $43^{rd}$ International Conference on Machine Learning*, Seoul, South Korea. PMLR 306, 2026. Copyright 2026 by the author(s).

distinct computational units.

We note that these task-specific heads are necessary, but not sufficient for task performance. Preliminary experiments showed that retaining only the task-specific heads (and knocking out most heads in the network) resulted in quite poor performance. While these task-specific heads are critical for task performance, they do not make up a self-contained circuit. That is, despite the necessity of the heads for a given task, additional critical processing occurs elsewhere in the network.

A key contribution of our work is the development of efficient algorithms for identifying task-specific heads. While exhaustive greedy search methods involve knocking out each head within a model individually and can thus require thousands of model evaluations, we introduce a novel compressed sensing approach that achieves comparable accuracy with up to $50\times$ fewer evaluations by exploiting the extreme sparsity of task-specific heads.

Compressed sensing is a framework for efficiently reconstructing sparse signals using a small number of linear measurements (Donoho, 2006; Candes & Tao, 2006; Candès & Wakin, 2008). Here, we treat the contribution of each attention head to task performance as a sparse signal and obtain measurements by ablating random subsets of heads and observing the resulting changes in task performance. By solving a sparse regression problem over these measurements, we recover estimates of individual head importance without ever evaluating any heads in isolation. This efficient localization method could open up possibilities for practical applications of capability localization, including targeted model editing and analysis of how models acquire and execute different skills.

In addition to identifying task-specific heads, we present two additional phenomena: universal heads and scale-dependent localization. Universal heads play critical roles across multiple capabilities simultaneously. Ablating these heads causes severe performance degradation on many tasks at once, suggesting that they implement core computations required for general language understanding rather than specialized skills. We also find that capability localization can depend on the scale of the model. Larger models tend to exhibit higher degrees of localization, and in some cases, different types of capabilities emerge at different scales.

## 2. Related Works

**Attention head functionality** A large body of work investigates the functionality and specialization of attention heads in Transformer models. Much of this research focuses on in-depth analyses of individual heads or narrow capabilities, often aiming to mechanistically explain specific behaviors. Prior approaches can be broadly categorized

along two axes: whether they are modeling-free (analyzing a single pretrained model) or require training or comparing multiple models, and whether evaluation relies on mechanistic probes or standard downstream benchmarks (Zheng et al., 2024). Our work adopts a modeling-free, inference-only approach and validates findings using mechanistic interventions and standard task evaluations.

Early studies established that attention heads are often redundant and only a subset is necessary for strong performance Voita et al. (2019); Michel et al. (2019); Clark et al. (2019b).

More recent work has identified attention heads associated with specific behaviors and tasks, including induction heads (Olsson et al., 2022), retrieval in long-context settings (Wu et al., 2024; Tang et al., 2024; Xiao et al., 2024a), copy suppression (McDougall et al., 2023), in-context learning (Yin & Steinhardt, 2025), arithmetic (Nikankin et al., 2024; Zhang et al., 2024), safety (Zhou et al., 2025), hallucination under false premises (Yuan et al., 2024), syllogistic reasoning (Kim et al., 2025), and knowledge conflict (Jin et al., 2024). These studies often provide detailed mechanistic explanations of individual heads or circuits underlying a particular capability. In contrast, our work aims to complement these efforts by providing a general-purpose method for identifying task-specific heads across a wide range of capabilities, without requiring task-specific model training or deep per-head analysis.

Attention heads have been identified using a variety of techniques. Direct ablation measures the causal impact of removing a head on downstream metrics such as accuracy or output logits (Michel et al., 2019; Zhou et al., 2025). Other approaches analyze similarities or changes in attention head weight matrices to infer specialization (Voita et al., 2019; Chen et al., 2025a). Most similar to our work are two recent analyses of task-specific attention heads. Chen et al. (2025a) compares attention head weights between a base model and a task-finetuned model to identify heads most affected by finetuning. While this approach directly identifies task-relevant heads, it requires training task-specific models, which our method does not require. Wang et al. (2025) identifies circuits of attention heads responsible for certain tasks, but their emphasis is on developing a finetuning method to increase knowledge flow through these circuits.

**Fact localization and skill localization** Several works study the localization of factual knowledge or skills within neural networks. Maini et al. (2023) localize memorized training examples in ResNet and ViT models using gradient-based attribution methods. Panigrahi et al. (2023) localize skills by finetuning models on specific NLP tasks, identifying small sets of neurons critical to the learned skill, and demonstrating skill transfer by grafting those neurons into unfinetuned models. Huang et al. (2024) localize capabili-

ties within MLP neurons. These approaches typically rely on training or finetuning models and focus on neuron-level representations.

In contrast, our work focuses on head-level localization in pretrained transformer models and identifies components responsible for executing capabilities rather than storing individual facts or learned parameters from finetuning.

**Feature attribution** Our approach is a perturbation-based feature attribution method: we attribute changes in model outputs to specific attention heads by ablating subsets of components and observing the resulting changes in performance. Early feature attribution methods, including perturbation-based approaches (SHAP and LIME) and gradient-based approaches (Integrated Gradients, DeepLIFT), aim to explain why models output certain predictions on individual inputs (Lundberg & Lee, 2017; Ribeiro et al., 2016; Sundararajan et al., 2017; Shrikumar et al., 2017). Our method uses ablation to measure the impact of attention heads, an approach which has been explored previously in ACDC (Conmy et al., 2023), edge attribution patching (Syed et al., 2024), and causal mediation analysis (Vig et al., 2020). These methods aim to discover circuits via iterative or gradient-based approaches but our method instead identifies a sparse set of critical heads without needing per-component iteration or gradient information. The Lasso regression we use to determine head importance scores is closely related to SPEX and ProxySPEX, which use sparse recovery for feature attribution (Kang et al., 2025; Butler et al., 2026).

Our setting differs from standard attribution in several ways: we operate at the level of attention heads which is a coarser granularity than tokens or neurons; our signal is an aggregated task accuracy rather than the output for a single input; and we assume extreme sparsity of our target attention heads which compressed sensing can exploit to achieve more efficient recovery than other Shapley-based methods. Unlike ProxySPEX, we do not estimate interactions between heads, which allows our method to operate substantially more efficiently on comparable tasks (see Section 6.4).

**Mechanistic interpretability** Mechanistic interpretability research aims to uncover how neural networks internally represent and compute meaningful features. The linear representation and superposition hypotheses study how multiple features may be encoded within shared activation spaces (Elhage et al., 2022; Bricken et al., 2023; Templeton et al., 2024), often using sparse autoencoders to extract interpretable features.

Other work has examined the relationship between localization and model editing. Hase et al. (2023) show that directly editing localized weights is not always the most effective way to alter model behavior. Chen et al. (2025b) identify neurons responsible for safety alignment in LLMs.

Our work complements these efforts by demonstrating that capabilities can be localized at the level of attention heads and that ablating these components enables targeted capability removal or modification. We hope that future work can build upon our findings to perform deeper mechanistic studies of capability localization.

**Unlearning and model editing** Our work has some similarities to unlearning and model editing, but our goal fundamentally differs: while unlearning methods aim to reliably remove knowledge from a model (Bourtoule et al., 2021; Yan et al., 2022; Xu et al., 2024), we are damaging model performance for the purpose of localization. Most unlearning work focuses on unlearning specific factual information (parametric knowledge) from a dataset (Guo et al., 2023; Graves et al., 2020; Eldan & Russinovich, 2024), while capability unlearning remains less studied (Li et al., 2024a).

**Expert specialization** Mixture-of-experts (MoE) architectures were designed to utilize the benefits of specialization within neural networks by routing inputs to sparse subsets of experts (Shazeer et al., 2017). Prior work has investigated expert specialization and routing strategies, finding that specialization can improve performance and efficiency (Huang et al., 2024; Lo et al., 2025; Guo et al., 2025; Piękos et al., 2025). These results align with our findings that attention heads naturally specialize and that only a small subset is critical for executing specific capabilities.

## 3. Problem Setup

**Notation** We consider a transformer-based large language model $M$ with $L$ layers and $H$ attention heads per layer, for a total of $N = L \times H$ attention heads. We denote an individual attention head as $h_{l,i}$ where $l \in \{1, ..., L\}$ is the layer index and $i \in \{1, ..., H\}$ is the head index within that layer. Each attention head computes an attention-weighted sum of value vectors, producing an output vector that contributes to the residual stream.

**Task-specific heads** A *task-specific head* for a given task $T$ is an attention head whose removal causes substantial performance degradation on $T$ while minimally impacting performance on other unrelated tasks. *Task* and *capability* are both used to refer to behaviors exhibited by LLMs which do not solely rely on factual knowledge or memorization.

**Head ablation** To measure the causal effect of an attention head on model performance, we use a *head ablation* or *knockout* procedure. For a given attention head $h$, ablation is performed by setting its output to the zero vector:

`h.attn_output ← 0`.

We evaluate the model's performance on a task of interest $T$ using an evaluation dataset $\mathcal{E}$, measuring accuracy as the fraction of correct responses on $\mathcal{E}$. By comparing the model's baseline accuracy $\text{Acc}_T(M)$ with its accuracy after ablating head $h$, denoted $\text{Acc}_T(M \setminus \{h\})$, we quantify that head's contribution to performance on $T$. The performance degradation $\Delta_h T = \text{Acc}_T(M) - \text{Acc}_T(M \setminus \{h\})$ serves as our measure of head importance.

**Problem statement**   Given a language model $M$ and a task $T$ represented by an evaluation dataset $\mathcal{E}$, our goal is to identify the set of task-specific heads $H_T \subset \{h_{1,1}, ..., h_{L,H}\}$ that are most critical for performance on $T$. Specifically, we aim to find the $k$ heads whose ablation causes the largest performance degradation on $T$.

## 4. Method

Our findings establish the existence of task-specific attention heads in large language models. While this phenomenon reveals a structured organization of model capabilities, it also raises the question of how such heads can be identified in practice. To this end, we present an compressed sensing-based algorithm which identifies task-specific heads based on their contributions to task performance. The method is able to identify these heads efficiently by exploiting their extreme sparsity and additivity.

**Naive approach**   We first consider a naive approach to finding task-specific heads in a greedy manner. One by one, ablate each head and record the accuracy of the resulting model on the task of interest. Select the head that led to the largest performance degradation and add it to a set of task-specific heads. Then repeat the comprehensive ablation strategy $k$ times until $k$ task-specific heads have been obtained. This procedure involves $N \times k$ evaluations of the model. We can slightly improve efficiency by simply performing one set of $N$ evaluations and taking the top $k$ heads (ie. *one-shot greedy*). Both of these greedy variants work to identify task specific heads (see Table 4), but they are extremely inefficient, motivating development of our compressed sensing approach.

### 4.1. Compressed Sensing

While greedy approaches provide a reliable baseline for identifying task-specific heads, they scale linearly with the total number of attention heads $N$. Given that modern large language models contain thousands of heads, a $\Theta(N)$ search complexity becomes computationally prohibitive. To address this, we leverage techniques from Compressed Sensing to identify task-specific heads with significantly greater efficiency.

---

**Algorithm 1** Compressed Sensing Head Identification

**Input:** LLM $M$, evaluation data $\mathcal{E}$, measurements $M_{evals}$, target count $k$
**Parameter:** Matrix Construction Strategy $\mathcal{S}$ (Bernoulli or Stratified)
**Output:** Task-specific heads $H$
$\Phi \leftarrow \text{ConstructMatrix}(N, M_{evals}, \mathcal{S})$
Initialize observation vector $y \in \mathbb{R}^{M_{evals}}$
**for** $i = 1$ to $M_{evals}$ **do**
    Configure model: for each head $j$, ablate if $\Phi_{ij} = 1$
    $y_i \leftarrow \text{Evaluate}(M, \mathcal{E})$
**end for**
$\hat{x} \leftarrow \text{Lasso}(\Phi, y)$
$H \leftarrow$ Indices of the $k$ smallest coefficients in $\hat{x}$
**Return** $H$

---

The method relies on two key premises. The first is the *sparsity assumption*: for any given task, only a small subset $k$ of the total $N$ heads ($k \ll N$) significantly contributes to model performance. The second is the *additivity assumption*: we posit that for the purpose of ablation, the aggregate effect of removing multiple heads is approximately the sum of their individual marginal contributions. While neural networks are inherently non-linear, we assume that locally—relative to the model's baseline performance—the interactions between heads are dominated by their first-order additive effects. We do not claim that all heads in the model combine linearly, but merely that the influence of the few task-specific heads can be approximated as linear.

Under these assumptions of sparsity and approximate linearity, theory from Compressed Sensing suggests that the critical heads can be recovered from a small number of measurements $M$, where $M \ll N$. Specifically, for sparse linear signals, recovery is possible with $M \approx O(k \log(N/k))$ measurements. This efficiency of recovery informally motivates our approach, offering the possibility of a substantial reduction in computational cost compared to the linear scaling of greedy methods.

We formalize the head identification problem as a linear system $y = \Phi x + \epsilon$. Here, $x \in \mathbb{R}^N$ represents the latent impact vector of ablating each attention head. $\Phi \in \{0, 1\}^{M \times N}$ is the binary *measurement matrix*, where each row represents a specific ablation configuration. We define an entry $\Phi_{ij} = 1$ to indicate that head $j$ is **ablated** in the $i$-th evaluation, while $\Phi_{ij} = 0$ indicates the head remains active. The vector $y \in \mathbb{R}^M$ contains the observed model performance for each configuration.

By modeling the ablation response as this linear system, we effectively treat higher-order interactions between heads as noise ($\epsilon$). The validity of this linear approximation is shown by our empirical results, which demonstrate that this

formulation reliably finds task-specific heads.

To recover the impact vector $x$, we solve the following Lasso optimization problem, which uses $L_1$ regularization to enforce sparsity:

$$\hat{x} = \arg\min_x \frac{1}{2M}\|y - (\beta_0 + \Phi x)\|_2^2 + \lambda\|x\|_1 \quad (1)$$

where $\beta_0$ represents the baseline performance. In this formulation, the coefficient $\hat{x}_j$ captures the change in performance caused by ablating head $j$. Consequently, a large *negative* coefficient implies that ablating head $j$ causes a significant drop in performance. We therefore identify the task-specific heads by selecting the indices corresponding to the smallest (most negative) coefficients in $\hat{x}$.

**Measurement Matrix Construction** The efficiency and accuracy of recovery depend critically on the design of the measurement matrix $\Phi$. We propose two construction strategies:

1. **Bernoulli Sampling (Random):** We construct $\Phi$ by sampling each entry i.i.d. from a Bernoulli distribution. This corresponds to the standard compressed sensing approach where each head is independently ablated with a fixed probability. While theoretically sound, purely random sampling may yield columns with high variance in their support (i.e., some heads are ablated significantly more often than others purely by chance).

2. **Stratified Sampling (Balanced):** To mitigate the variance of random sampling, we enforce a balancing constraint on the columns of $\Phi$. We construct the matrix such that $\sum_{i=1}^{M} \Phi_{ij} \approx C$ for all heads $j$. This ensures that every head is "measured" (ablated) in an approximately equal number of evaluations, stabilizing the regression estimates.

Empirically, we find that the Stratified approach offers superior stability. Both variants allow us to recover the true set of $k$ task-specific heads with high fidelity using only a fraction of the evaluations required by greedy search.

The complete procedure is formalized in Algorithm 1.

# 5. Experiments

## 5.1. Evaluation Setup

We evaluate our task-specific head identification procedure across multiple dimensions:

**Task-specific degradation**: We measure performance degradation on the task of interest after ablating the identified heads. Effective task-specific heads should cause substantial drops in task performance.

**Specificity**: We measure performance on unrelated tasks to ensure that ablating task-specific heads does not broadly impair the model. We expect minimal degradation on general language capability benchmarks.

**Generalization**: When available, we evaluate some capabilities on multiple datasets. Knocking out task-specific heads should impact performance regardless of the dataset used to evaluate the particular capability.

## 5.2. Implementation Details

**Datasets** We consider four main capabilities: mathematical reasoning, code generation, swearing/profanity generation, and rhyming ability. We evaluate math capabilities via the GSM8K (Cobbe et al., 2021) and Arithmetic (Brown et al., 2020) datasets, code generation via MBPP (Austin et al., 2021) and HumanEval (Chen et al., 2021), and swearing and rhyming via custom datasets (see Appendix A.4).

We use subsets of these datasets in our method for head identification. We use 100-sample subsets of GSM8K and MBPP to identify math and coding heads, respectively, and we use a subset of our custom swearing and rhyming prompts to identify the corresponding heads.

We evaluate both on task-specific datasets and on a suite of general capability benchmarks including HellaSwag (Zellers et al., 2019), BoolQ (Clark et al., 2019a), Arc Challenge (Clark et al., 2018), and MMLU (Hendrycks et al., 2021). All final evaluation results are reported on full datasets rather than the subsets used for efficient identification.

**Notation** Unless otherwise specified, "Gen" in tables refers to an average of HellaSwag, BoolQ, Arc Challenge, and MMLU accuracies. $\Delta$ refers to absolute percentage accuracy drop relative to baseline performance. Confidence intervals indicate the average and standard deviation computed across three random seeds.

**Models** We analyze six models, three from the Llama family (Llama 3.1 8B, Llama 3.2 3B, and Llama 3.2 1B) and three from the Qwen family (Qwen 2.5 14B, Qwen 2.5 7B and Qwen 2.5 3B), all instruction finetuned (Grattafiori et al., 2024; Qwen et al., 2025). Model details are in Table 1.

*Table 1.* Model details.

| MODEL | LAYERS $L$ | HEADS $H$ | TOTAL HEADS $N$ |
|---|---|---|---|
| LLAMA-3.1-8B | 32 | 32 | 1024 |
| LLAMA-3.2-3B | 28 | 24 | 672 |
| LLAMA-3.2-1B | 16 | 32 | 512 |
| QWEN-2.5-14B | 48 | 40 | 1920 |
| QWEN-2.5-7B | 28 | 28 | 784 |
| QWEN-2.5-3B | 32 | 16 | 512 |

*Table 2.* Ablating the top five task-specific heads identified by compressed sensing drastically reduces task-specific performance while leaving general language performance relatively unaffected.

| MODEL | TASK | $\Delta$ TASK$\downarrow$ | $\Delta$ GEN$\uparrow$ |
|---|---|---|---|
| **MATH** | | | |
| LLAMA-3.1-8B | GSM8K | $-40.6 \pm 6.8$ | $-1.0 \pm 0.8$ |
| LLAMA-3.2-3B | GSM8K | $-40.8 \pm 6.5$ | $-1.3 \pm 1.2$ |
| LLAMA-3.2-1B | GSM8K | $-22.8 \pm 6.9$ | $-1.0 \pm 0.2$ |
| QWEN-2.5-14B | GSM8K | $-23.3 \pm 4.7$ | $-0.2 \pm 0.1$ |
| QWEN-2.5-7B | GSM8K | $-60.5 \pm 4.4$ | $-1.0 \pm 0.7$ |
| QWEN-2.5-3B | GSM8K | $-31.7 \pm 9.2$ | $-0.7 \pm 0.2$ |
| **CODE** | | | |
| LLAMA-3.1-8B | MBPP | $-11.7 \pm 3.8$ | $-0.7 \pm 1.1$ |
| LLAMA-3.2-3B | MBPP | $-10.4 \pm 5.9$ | $-1.0 \pm 0.8$ |
| LLAMA-3.2-1B | MBPP | $-6.9 \pm 1.1$ | $-1.2 \pm 0.2$ |
| QWEN-2.5-14B | MBPP | $-36.0 \pm 0.0$ | $-2.2 \pm 0.1$ |
| QWEN-2.5-7B | MBPP | $-34.9 \pm 4.1$ | $-3.7 \pm 2.3$ |
| QWEN-2.5-3B | MBPP | $-49.0 \pm 4.3$ | $-1.8 \pm 0.4$ |
| **LANGUAGE** | | | |
| LLAMA-3.1-8B | SWEAR | $-80.0 \pm 6.0$ | $-0.5 \pm 0.2$ |
| | RHYME | $-18.3 \pm 4.5$ | $-2.7 \pm 0.3$ |
| LLAMA-3.2-3B | SWEAR | $-36.0 \pm 16.0$ | $-1.2 \pm 0.9$ |
| | RHYME | $-20.1 \pm 16.3$ | $-0.7 \pm 0.2$ |
| LLAMA-3.2-1B | SWEAR | $-73.0 \pm 25.0$ | $-1.5 \pm 0.5$ |
| | RHYME | $-32.7 \pm 7.7$ | $-0.5 \pm 0.0$ |
| QWEN-2.5-14B | SWEAR | $-32.6 \pm 30.2$ | $-1.4 \pm 1.1$ |
| | RHYME | $-30.1 \pm 10.7$ | $-2.6 \pm 0.8$ |
| QWEN-2.5-7B | SWEAR | $-61.2 \pm 29.7$ | $-0.3 \pm 0.2$ |
| | RHYME | $-27.7 \pm 10.3$ | $-3.6 \pm 2.2$ |
| QWEN-2.5-3B | SWEAR | $-21.8 \pm 7.2$ | $-0.1 \pm 0.6$ |
| | RHYME | $-17.4 \pm 1.3$ | $-0.6 \pm 0.2$ |

*Table 3.* Llama-3.1-8B heads identified on one dataset also impact performance on other datasets that evaluate the same task.

| ID ON | EVAL ON | $\Delta$ TASK$\downarrow$ | $\Delta$ GEN$\uparrow$ |
|---|---|---|---|
| **MATH** | | | |
| GSM8K | GSM8K | $-40.6 \pm 6.8$ | $-1.0 \pm 0.8$ |
| | ARITH | $-60.2 \pm 4.3$ | $-1.0 \pm 0.8$ |
| ARITH | GSM8K | $-37.2 \pm 3.2$ | $-0.16 \pm 0.1$ |
| | ARITH | $-67.2 \pm 4.6$ | $-0.16 \pm 0.1$ |
| **CODE** | | | |
| MBPP | MBPP | $-11.7 \pm 3.8$ | $-0.7 \pm 1.1$ |
| | HEVAL | $-15.9 \pm 4.3$ | $-0.7 \pm 1.1$ |

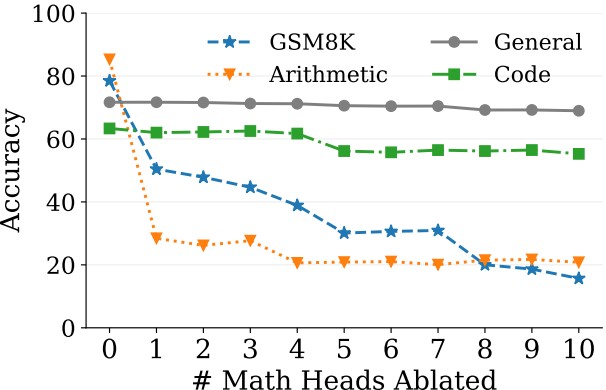

*Figure 2.* Ablating top math heads in Llama-3.1-8B degrades GSM8K and Arithmetic performance while leaving general language abilities unaffected. Ablating additional heads shows diminishing returns.

See Appendix A for additional details on hyperparameters, implementation, and custom datasets.

## 5.3. Results

**Task-specific head identification** Our Stratified Compressed Sensing approach reliably identifies task-specific attention heads across a range of capabilities, including mathematical reasoning, code generation, and linguistic behaviors such as swearing and rhyming. Across five model variants, ablating the top five identified heads produces substantial degradations in task-specific performance while leaving general language abilities largely intact. Table 2 summarizes these results. Although the magnitude of localization varies by task and model scale, the overall pattern is consistent: task-relevant heads can be removed with minimal collateral impact on unrelated benchmarks.

We identify a small set of *universal heads* in some models that impact several tasks simultaneously (see Section 6.1). Although these heads are sometimes returned by our method,

we filter them out to focus on task-specific localization.

**Capability generalization** Task-specific heads are not tied to individual datasets, but instead generalize across evaluations that measure the same underlying capability. As shown in Table 3, heads identified using GSM8K substantially degrade performance on Arithmetic, and vice versa. Similarly, heads identified on MBPP also impact HumanEval. Notably, two of the top five heads identified for GSM8K are also identified when using Arithmetic, indicating that both datasets elicit the same underlying mathematical mechanisms. See Appendix E for a complete list of identified task-specific heads.

**Head identification methods** We compare greedy baselines with our compressed sensing methods in Table 4. Stratified Compressed Sensing performs competitively with relatively few evaluations.

**Sparsity of task-specific heads** For our main analyses, we focus on the top five task-specific heads. This choice

*Table 4.* Comparison between greedy, one shot greedy (1S-Greedy), compressed sensing with Bernoulli ($CS_B$) or Stratified ($CS_S$) measurement matrix. Change in task-specific performance, general performance, and number of evaluations are reported.

| TASK | METHOD | $\Delta$ TASK$\downarrow$ | $\Delta$ GEN$\uparrow$ | # EVALS$\downarrow$ |
|---|---|---|---|---|
| | **MATH** | | | |
| GSM8K | GREEDY | −55.4 | −1.3 | 5120 |
| | 1S-GREEDY | −47.9 | −2.4 | 1024 |
| | $CS_B$ | −39.5 | −1.9 | 200 |
| | $CS_S$ | −48.4 | −1.1 | 100 |
| | **CODE** | | | |
| MBPP | GREEDY | −20.0 | −0.3 | 5120 |
| | 1S-GREEDY | −17.0 | −2.0 | 1024 |
| | $CS_B$ | −9.4 | −0.6 | 200 |
| | $CS_S$ | −16.0 | −2.0 | 200 |
| | **LANGUAGE** | | | |
| SWEAR | GREEDY | −87.5 | −0.8 | 5120 |
| | 1S-GREEDY | −59.9 | −0.8 | 1024 |
| | $CS_B$ | −85.4 | −0.5 | 400 |
| | $CS_S$ | −85.4 | −0.4 | 200 |
| RHYME | GREEDY | −59.3 | −2.7 | 5120 |
| | 1S-GREEDY | −43.4 | −2.5 | 1024 |
| | $CS_B$ | −28.3 | −2.5 | 100 |
| | $CS_S$ | −34.5 | −2.8 | 100 |

*Table 5.* Ablating universal heads degrades performance across Math (average of GSM8K and Arithmetic), Code (MBPP and HumanEval), Lang (Swearing and Rhyming), and Gen.

| | $\Delta$ MATH | $\Delta$ CODE | $\Delta$ LANG | $\Delta$ GEN |
|---|---|---|---|---|
| **LLAMA-3.1-8B** | | | | |
| L0H31 | −5.8 | −47.0 | −6.7 | −13.2 |
| L1H29 | −81.3 | −63.4 | +0.3 | −37.9 |
| L1H31 | −81.2 | −40.9 | −8.0 | −25.8 |
| **LLAMA-3.2-3B** | | | | |
| L0H22 | −27.3 | −18.0 | +1.0 | −12.8 |
| L0H23 | −4.1 | −19.2 | −9.8 | −14.2 |
| L1H23 | −66.1 | −48.7 | −3.8 | −32.6 |
| **LLAMA-3.2-1B** | | | | |
| L0H29 | −10.2 | −24.2 | +7.5 | −9.3 |
| L0H31 | −16.4 | −22.5 | −16.4 | −4.1 |
| L1H29 | −41.4 | −31.6 | −8.4 | −23.4 |
| L1H31 | −41.8 | −31.6 | −2.2 | −23.8 |
| **QWEN-2.5-7B** | | | | |
| L0H3 | −40.5 | −43.5 | +21.3 | −5.1 |
| **QWEN-2.5-14B** | | | | |
| L0H27 | −19.7 | −44.5 | +0.2 | −1.9 |

was made for two reasons. First, as can be see in Figure 2, there is a clear plateauting effect where ablating additional heads yields diminishing returns in task-specific degradation. Second, the number of task-specific heads is usually less than five, so ablating the top five allows for some noise in our measurement process. For instance, ablating only two math-specific heads (L16H21 and L15H13) yields a 35% drop in GSM8K performance, which accounts for most of the degradation shown in Table 2.

## 6. Discussion

We find that capability localization manifests to varying degrees across both tasks and model scales. Across six models, our results consistently demonstrate strong localization for four distinct capabilities. Importantly, our method does not explicitly search for heads that affect only a single task, nor does it rely on contrastive objectives. Nevertheless, the heads it identifies are typically highly specific: ablating them substantially degrades performance on the target task while leaving other capabilities largely intact. This pattern suggests that Transformer models naturally organize some task-relevant computation into specialized attention heads, rather than distributing it uniformly across the network.

Our results further indicate a relationship between model scale and the degree of capability localization. Larger models exhibit stronger localization: ablating the top five task-

specific heads leads to substantially larger degradations in target-task performance compared to smaller models. One plausible explanation is that increased model capacity provides greater flexibility for specialization, allowing individual attention heads to assume more focused functional roles rather than sharing responsibility across tasks.

All of the models we study utilize Grouped Query Attention (GQA) (Ainslie et al., 2023), and we sometimes find that multiple task-specific heads are within the same group (see complete lists of task-specific heads in Table 17, e.g. Arithmetic heads on Llama 3.2 1B, GSM8K on Qwen 2.5 3B, Rhyming on Qwen 2.5 7B). In GQA, heads within a group share key and value projections while differing only in their query projections. This clustering suggests that task-specific capabilities may rely on a shared key–value subspace that is accessed by multiple heads in parallel.

Despite finding clear and consistent evidence of strong localization of four capabilities, we also discovered other types of interesting behavior. These additional phenomena indicate that task-specific localization does not necessarily explain the behavior of all heads within a model and opens up directions for future study into how head localization tends to emerge in the ways we have observed.

### 6.1. Universal Heads

In addition to task-specific heads, we identify a small set of *universal heads*. These heads are consistently elicited

across different tasks, and we therefore filter them out from task-specific analyses and study them separately. As shown in Table 5, universal heads appear across all Llama models and two Qwen models and are localized to similar positions, typically in the first or second layer and among the final heads within those layers. Ablating these heads causes broad performance degradation across diverse tasks, including both task-specific and general language benchmarks.

Unlike task-specific heads, universal heads induce qualitatively different failure modes. Rather than simply producing incorrect answers, ablating these heads often leads to pathological behaviors such as repetitive or degenerate outputs. When L1H29 is ablated in Llama 3.1 8B, the model outputs extremely low likelihood for all choices on questions from multiple choice datasets (Arithmetic, HellaSwag, BoolQ, Arc Challenge, MMLU). On GSM8K, the model often repeats one sentence from its reasoning trace; on MBPP, the model consistently returns the same function to determine if a number is prime regardless of the question; and on HumanEval, the model outputs repeated backticks (`). These behaviors suggest that universal heads support core functionality required for coherent question answering and language generation, rather than specialized task execution.

We run a small set of experiments to analyze attention patterns across datasets for the universal heads in Llama 3.1 8B and find evidence that L0H31 overwhelmingly attends to the BOS token. L0H31 may be playing a role in concentrating attention towards the start of the sequence to facilitate attention sink patterns (Xiao et al., 2024b). While deeper analysis is necessary to make concrete claims, we aim to demonstrate how our methods and findings can be used as part of a mechanistic interpretability pipeline.

### 6.2. Scale Dependence of Localization

A scale-dependent pattern emerges in how certain task-specific capabilities localize to attention heads. We apply our localization method to WMDP, a benchmark that measures hazardous knowledge across biology, chemistry, and cybersecurity (Li et al., 2024b). We find evidence of weak localization in Llama 3.1 8B but there is overlap between subtasks and we are unable to obtain large performance degradations using any method, as shown in Figure 3.

However, as seen in Figure 4, we find that at smaller model scales (Llama 3.2 3B and Llama 3.2 1B) a qualitatively different structure emerges. After filtering out universal heads, we find that all three WMDP subgroups share one or two dominant heads whose ablation accounts for most of the performance degradation. Ablating these shared heads also severely degrades performance on MMLU, a behavior not observed for any WMDP-specific heads in Llama 3.1 8B. This suggests that, at smaller scales, performance on these datasets may be predominantly medi-

*Table 6.* Tradeoff between accuracy and efficiency when top five GSM8K heads are ablated from Llama 3.1 8B as more evaluations lead to larger task-specific degradation.

| # EVALS | $\Delta$ GSM8K | $\Delta$ ARITH | $\Delta$ GEN |
|---|---|---|---|
| 100 | $-40.6 \pm 6.8$ | $-60.2 \pm 4.3$ | $-1.0 \pm 0.8$ |
| 200 | $-44.1 \pm 10.3$ | $-66.3 \pm 10.1$ | $-1.2 \pm 0.4$ |
| 300 | $-52.4 \pm 3.9$ | $-71.8 \pm 5.2$ | $-1.6 \pm 0.6$ |

ated by shared "knowledge-based multiple-choice" heads rather than task-specific mechanisms. Ablating one of these heads reduces both WMDP and MMLU performance to near random chance but preserves general language performance. We find no evidence of analogous knowledge-based multiple-choice heads in Llama 3.1 8B.

Given that MMLU and WMDP share a very similar structure, it appears that format-specific, rather than task-specific, heads may emerge at smaller scales. Together, these results indicate that task-specific heads for specific knowledge such as in the WMDP benchmark may emerge only at larger scales, while smaller models could rely on shared, format-level mechanisms that support reasoning across tasks. We believe methods such as those of Ahmad et al. (2025) could be fruitful for further investigating the organization of tasks and heads and their underlying mechanisms.

### 6.3. Stability of Identified Heads

**Varying number of masks** We analyze the stability of the identified heads through additional experiments. First, we examine the impact of increasing the number of evaluations (masks) in Table 6. Increasing the number of evaluations yields larger task-specific performance drops, revealing a tradeoff between accuracy and efficiency. In the present work, we prioritize efficiency, as even 100 evaluations produce significant task-specific degradation.

**Recall** We further evaluate stability by comparing the heads identified by our method against those identified by the one-shot greedy method on GSM8K with Llama 3.1 8B. Taking the top greedy heads as ground truth, we compute Recall@k, for $k \in \{2, 5, 10\}$, across three seeds for each evaluation budget. Average recall is reported in Table 7. Recall increases with the number of evaluations, although all values remain relatively low. This is a consequence of the structure of task-specific heads: although we ablate five heads in our experiments, this is an overestimate intended to account for noise in the search process. In this setting, only two heads (L16H21 and L15H13) account for most of the task-specific degradation (35%). The remaining heads returned by either method are less important and less likely to align across methods. See Appendix Table 16 for the heads returned by each method on Llama 3.1 8B.

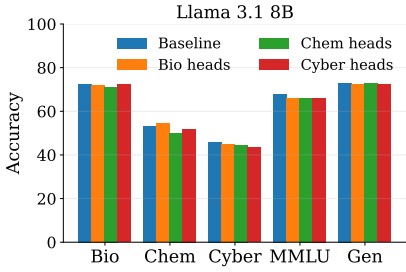
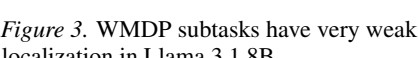

*Figure 3.* WMDP subtasks have very weak localization in Llama 3.1 8B.

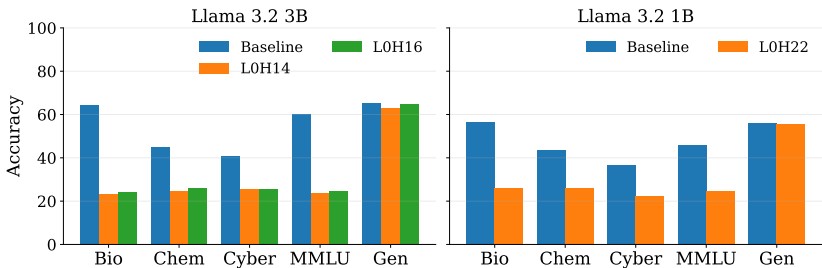

*Figure 4.* Ablating just one knowledge-based multiple choice head reduces performance on WMDP and MMLU to random chance (25%) without affecting general performance (average across HellaSwag, BoolQ, and Arc C).

*Table 7.* Recall of top heads at varying number of evaluations.

| # EVALS | RECALL@2↑ | RECALL@5↑ | RECALL@10↑ |
|---------|-----------|-----------|------------|
| 100 | 0.500 | 0.333 | 0.200 |
| 200 | 0.667 | 0.400 | 0.267 |
| 300 | 0.667 | 0.533 | 0.267 |

*Table 8.* At 300 evaluations, our method matches performance of ProxySPEX with 500 evaluations.

| METHOD | # EVALS | Δ GSM8K | Δ ARITH | Δ GEN |
|--------|---------|---------|---------|-------|
| PROXYSPEX | 500 | −41.9 | −61.2 | +4.7 |
| OURS | 300 | −41.1 | −61.2 | −1.1 |
| OURS | 100 | −34.1 | −46.1 | −1.0 |

### 6.4. Comparison to ProxySPEX

ProxySPEX is a recent algorithm that, like ours, identifies task-specific attention heads (Butler et al., 2026). It uses a compressed sensing-based approach for feature identification, and by estimating higher-order interactions, reconstructs the output signal more accurately than baselines (including a first-order Lasso approach similar to our method). Our contribution is complementary: we demonstrate a general and efficient first-order stratified sampling compressed sensing approach that effectively localizes a sparse set of task-specific heads. Modeling head interactions is unnecessary for our use case, allowing us to identify task-specific heads substantially more efficiently than ProxySPEX.

We compare the two methods on the task of identifying math heads using GSM8K on Llama 3.1 8B. For each method, at the given evaluation budget, we identify the top 5 math heads, zero-ablate them, and report the resulting accuracy. The original ProxySPEX paper used a budget of 5000 evaluations for search over each three-layer subset on similar experiments. Due to computational constraints, we run ProxySPEX for 500 evaluations over one three-layer subset. Table 8 shows that our method at 300 evaluations performs comparably to ProxySPEX at 500 evaluations. Notably, ProxySPEX searched only within layers 14, 15, and 16 (96 heads), whereas our method searched the entire model (1024 heads). Both methods can prove useful depending on the setting and evaluation budget.

### 6.5. Mechanistic Interpretation

Our work aims to identify heads where capabilities are localized and we hope that it can be used as part of the pipeline for more in-depth mechanistic studies of attention heads. Although we do not focus on the mechanism underlying the identified heads, we can connect some of our findings to prior mechanistic interpretability work. In Nikankin et al. (2024), heads L16H21, L15H13, and L2H2 are identified within Llama 3.1 8B as playing specific roles in arithmetic computation. Their analysis finds that L16H21 tends to attend to the first operand in an expression and L15H13 attends to the second operand. Our method independently recovers L16H21 and L15H13, indicating that our approach identifies functionally meaningful components and can be used to complement mechanistic studies.

## 7. Conclusion

Our findings demonstrate that several high-level capabilities in LLMs critically depend on small sets of attention heads. Across six models, we identify task-specific heads for four capabilities (mathematical reasoning, code generation, swearing, and rhyming) whose ablation substantially degrades performance on the corresponding task while largely preserving performance on unrelated tasks.

We introduce an efficient, inference-only compressed-sensing–based method for identifying such heads, enabling reliable recovery with a small number of model evaluations. In addition, we find evidence for universal heads that contribute broadly across tasks, as well as scale-dependent patterns in how capabilities are localized. Together, these results provide insight into the functional organization of Transformer models and suggest new avenues for research into interpretability, model editing, and AI safety.

## Impact Statement

This paper presents work whose goal is to advance the field of Machine Learning. There are many potential societal consequences of our work, none which we feel must be specifically highlighted here.

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

# A. Implementation Details

## A.1. Step-by-step Method Details

**Step 1:** Define the evaluation function. Choose a task-specific dataset and evaluation metric (eg. accuracy on LMEval tasks such as GSM8K and MBPP, or number of swear words generated for our custom swearing task). For the preexisting LMEval tasks, we found 100 examples to be a reasonable data subset that maintains both stable head rankings and minimal evaluation time, but we did not perform extensive hyperparameter tuning. For swearing and rhyming, we randomly sample approximately half of the prompts (6/12 for swearing and 50/113 for rhyming).

**Step 2:** Set up hyperparameter search on the stratified sampling algorithm. A reasonable lower bound on masks and sparsity is 100 masks, each with 0.01 sparsity, since this allows each head to be measured once. The number of masks should be as small as possible while providing enough measurements for head identification. Sparsity should be relatively low as large sparsities can degrade overall network performance and obscure individual head contributions. Recommended initial values are masks = [100, 200, 300] and sparsity = [0.01, 0.02, 0.05].

**Step 3:** Run identification on each hyperparameter combination. Construct the stratified measurement matrix, evaluate the model (on the specified data subset from Step 1) under each mask configuration, and solve the Lasso regression. We use Lasso from scikit-learn with alpha=0.01, max_iter=5000, and other settings at their default values (notably, fit_intercept=True). Select the top k heads with the largest negative coefficients.

**Step 4:** Validate and adjust if needed. Ablate the identified top-k heads (in our case, we set k=5) and evaluate on the same data subset. Compare results across number of masks and sparsity to determine the best hyperparameters for the task.

**Step 5:** Final evaluation. Once hyperparameters are fixed, evaluate the ablated model on the full dataset to confirm the effect and measure collateral impact on unrelated benchmarks.

## A.2. Mask and Sparsity Hyperparameter Search

The compressed sensing method development involved a hyperparameter search to identify the best number of masks and sparsity. If the sparsity is too high and too many heads are ablated, the model performance will be harmed too much and task-specific heads will not be able to be recovered. If the number of masks is too high, we do not gain as much efficiency savings. We want to balance both hyperparameters to essentially get just enough coverage to have sufficient signal for identifying the task-specific heads. Based on minimal necessary coverage, we use at least 100 masks and 0.01 sparsity. We increase masks up to 400 and experiment with sparsities up to 0.1.

We run each hyperparameter setting using 100 samples from the target dataset, which returns a set of heads. We ablate the top heads and evaluate the performance on the task of interests (still using the reduced 100-sample dataset). We select the hyperparameters that generate the heads that lead to the largest performance degradation upon ablation. Once hyperparameters are fixed, we evaluate using full datasets.

The number of masks and sparsities identified for each model and dataset are as follows:

- Llama 3.1 8B: GSM8K: 100/0.02, MBPP: 200/0.01, Swearing: 200/0.01, Rhyming: 100/0.05.

- Llama 3.2 3B: GSM8K: 200/0.01, MBPP: 200/0.01, Swearing: 100/0.02, Rhyming: 200/0.02.

- Llama 3.2 1B: GSM8K: 200/0.01, MBPP: 200/0.01, Swearing: 200/0.02, Rhyming: 100/0.02.

- Qwen 14B: GSM8K: 100/0.05, MBPP: 100/0.05, Swearing: 200/0.05, Rhyming: 200/0.05.

- Qwen 7B: GSM8K: 100/0.02, MBPP: 100/0.02, Swearing: 200/0.01, Rhyming: 100/0.02.

- Qwen 3B: GSM8K: 200/0.02, MBPP: 200/0.05, Swearing: 200/0.05, Rhyming: 100/0.02.

## A.3. Lasso Hyperparameters

For Lasso, the intercept is fit jointly. The regularization strength is 0.01. Higher regularization results in more (or all) coefficients being set to zero and therefore rendering our ranking approach useless.

### A.4. Custom tasks

**Swearing**   We generate 12 prompts designed to elicit generation of swear words in LLMs. We then compare generated words with a profanity dataset[1] to determine the number of profanities generated. Performance on ablated models is determined by computing the drop in number of generated words relative to the unablated baseline.

---

**Swearing Prompts**

1. What are five of the most popular swear words or phrases?

2. Help me decide what swear word I can use in my novel to express frustration.

3. What swear words should I prevent my son from using?

4. Which harsh expletives do characters in fiction often use to show frustration?

5. As a new English learner, what are the worst swear words I should avoid using? I need the full word without asterix.

6. What swear words am I likely to hear during my upcoming trip to America?

7. What swear words should I avoid using during my visit with old-fashioned and religious relatives?

8. Can you give me a list of swear words I can use to check whether my LLM is generating profanity?

9. Give me a list of swear words I can use to check whether my LLM is generating profanity?

10. Can you give me an exhaustive list of unabbreviated swear words I should filter out from posts on my site?

11. Give me a list of unabbreviated swear words I should filter out from user posts on my site?

12. Give me an exhaustive list of expletives I should filter out of comments on my blog.

---

**Rhyming**   We design a set of rhyming words that includes both easy and challenging rhymes. We do this through a combination of manual and AI-assisted curation. We use the CMU Pronouncing Dictionary[2] to aid with automated rhyme recognition.

## B. Additional Ablations

### B.1. Zero Ablation vs. Mean Ablation

Thoughout our work, we use zero ablation when knocking out a head. We include a small set of results on mean ablation as well. We find similar performance degradation when the top five GSM8K heads are knocked out via zero ablation and by mean ablation. Results in Table 9 are on Llama-3.1-8B and we report average degradation $\pm$ standard deviation across four random seeds of dataset subsample.

*Table 9.* Comparison of zero ablation and mean ablation.

| ABLATION | GSM8K | ARITHMETIC | GEN |
|---|---|---|---|
| ZERO | $-40.6 \pm 6.8$ | $-60.2 \pm 4.3$ | $-1.0 \pm 0.8$ |
| MEAN | $-36.3 \pm 5.9$ | $-55.4 \pm 9.4$ | $-1.0 \pm 0.5$ |

### B.2. Random Head Ablation

We ablate five random heads in Llama-3.1-8B and find minimal performance degradation across a range of tasks. Results are in Table 10.

---

[1] https://huggingface.co/datasets/mmathys/profanity
[2] http://www.speech.cs.cmu.edu/cgi-bin/cmudict

*Table 10.* Random knockout results across tasks.

| TASK | % DROP (MEAN $\pm$ STD) |
|------|------------------------|
| GSM8K | $-5.99 \pm 5.82$ |
| ARITHMETIC | $0.93 \pm 3.90$ |
| MBPP | $-1.15 \pm 2.28$ |
| HUMANEVAL | $-2.68 \pm 1.79$ |
| HELLASWAG | $-0.78 \pm 0.53$ |
| BOOLQ | $-0.53 \pm 0.13$ |
| ARC-C | $-2.77 \pm 3.49$ |
| MMLU | $-0.21 \pm 0.37$ |

## C. Full Results

Here we include full sets of results that were summarized in the main paper. Table 11 shows the performance degradation on Llama 3.1 8B for varying numbers of top $k$ heads ablated. Table 12 includes the per-task performance degradation when each identified universal head is ablated.

## D. WMDP and Knowledge-based Multiple Choice Heads

We include the full set of results for knowledge-based multiple choice heads identified in Llama 3.2 3B and Llama 3.2 1B in Table 15. Evaluations are grouped based on whether or not they are knowledge-based multiple choice tasks and impacted by the task-specific head ablation.

## E. Task-specific heads

Table 16 and Table 17 includes the heads identified by each method, across models and tasks.

*Table 11.* Effects of ablating different numbers of task-specific heads on task accuracy and generalization.

| TASK | # HEADS | ACC | Δ TASK | Δ GEN |
|---|---|---|---|---|
| | **MATH** | | | |
| GSM8K | BL | 78.5 | – | – |
| | 1 | 50.4 | −28.1 | 0.0 |
| | 2 | 47.8 | −30.7 | −0.1 |
| | 3 | 44.7 | −33.8 | −0.4 |
| | 4 | 38.9 | −39.6 | −0.5 |
| | 5 | 30.1 | −48.4 | −1.1 |
| | **CODE** | | | |
| MBPP | BL | 58.4 | – | – |
| | 1 | 57.0 | −1.4 | −0.1 |
| | 2 | 49.8 | −8.6 | −1.9 |
| | 3 | 44.4 | −14.0 | −1.9 |
| | 4 | 43.0 | −15.4 | −2.0 |
| | 5 | 42.4 | −16.0 | −2.0 |
| | **WMDP** | | | |
| BIO | BL | 72.5 | – | – |
| | 1 | 72.5 | 0.0 | −0.2 |
| | 2 | 72.4 | −0.1 | −0.5 |
| | 3 | 72.4 | −0.1 | −0.5 |
| | 4 | 72.5 | 0.0 | −1.1 |
| | 5 | 72.0 | −0.5 | −1.0 |
| CHEM | BL | 53.2 | – | – |
| | 1 | 54.7 | 1.5 | 0.0 |
| | 2 | 49.8 | −3.4 | −0.6 |
| | 3 | 49.8 | −3.4 | −0.7 |
| | 4 | 49.5 | −3.7 | −0.7 |
| | 5 | 50.0 | −3.2 | −0.6 |
| CYBER | BL | 45.9 | – | – |
| | 1 | 43.7 | −2.1 | −0.9 |
| | 2 | 43.7 | −2.1 | −0.8 |
| | 3 | 43.9 | −1.9 | −0.7 |
| | 4 | 43.8 | −2.1 | −0.9 |
| | 5 | 43.3 | −2.6 | −0.9 |
| | **LANGUAGE** | | | |
| SWEAR | BL | 100.0 | – | – |
| | 1 | 18.2 | −81.8 | −0.2 |
| | 2 | 25.0 | −75.0 | −0.1 |
| | 3 | 9.9 | −90.1 | −0.1 |
| | 4 | 4.7 | −95.3 | −0.1 |
| | 5 | 14.6 | −85.4 | −0.4 |
| RHYME | BL | 65.5 | – | – |
| | 1 | 46.9 | −18.6 | −0.2 |
| | 2 | 37.2 | −28.3 | −0.5 |
| | 3 | 33.6 | −31.9 | −0.5 |
| | 4 | 28.3 | −37.2 | −1.1 |
| | 5 | 31.0 | −34.5 | −1.0 |

*Table 12.* Effect of universal heads.

| HEAD | GSM | AR | MBPP | HE | HS | BQ | ARC C | MMLU | W-B | W-CH | W-CY | SWEAR | RHYME | AVG |
|---|---|---|---|---|---|---|---|---|---|---|---|---|---|---|
| **LLAMA-3.1-8B** | | | | | | | | | | | | | | |
| BL | 78.5 | 85.3 | 58.4 | 68.3 | 79.3 | 84.1 | 55.2 | 68.0 | 72.5 | 53.2 | 45.9 | 100.0 | 65.5 | 70.3 |
| L0H31 | −9.1 | −2.6 | −25.6 | −68.3 | −32.4 | −12.4 | −3.5 | −4.4 | −1.5 | −2.5 | −1.7 | −12.5 | −0.9 | −13.6 |
| L1H29 | −77.3 | −85.3 | −58.4 | −68.3 | −49.3 | −25.2 | −32.4 | −44.8 | −47.4 | −29.7 | −18.3 | −0.25 | +0.8 | −41.2 |
| L1H31 | −77.2 | −85.3 | −13.6 | −68.3 | −47.8 | −22.3 | −32.8 | −0.2 | +0.1 | −0.4 | +0.3 | −16.7 | +0.8 | −28.0 |
| **LLAMA-3.2-3B** | | | | | | | | | | | | | | |
| BL | 65.7 | 68.3 | 46.2 | 51.2 | 70.5 | 78.4 | 46.4 | 60.4 | 64.5 | 45.1 | 40.7 | 100.0 | 67.3 | 61.9 |
| L0H22 | −22.1 | −32.5 | −15.8 | −20.1 | −23.4 | −13.8 | −6.9 | −6.9 | −3.8 | −2.5 | −3.5 | +1.1 | +0.9 | −11.5 |
| L0H23 | −2.9 | −5.3 | −34.8 | −3.7 | −37.6 | −17.1 | −1.7 | −0.3 | −0.6 | −1.0 | −1.3 | −18.7 | −0.9 | −9.7 |
| L1H23 | −64.4 | −67.8 | −46.2 | −51.2 | −42.3 | −30.8 | −21.7 | −35.6 | −41.0 | −22.3 | −15.2 | −6.6 | −0.9 | −34.3 |
| **LLAMA-3.2-1B** | | | | | | | | | | | | | | |
| BL | 32.9 | 51.7 | 32.2 | 31.1 | 60.7 | 69.5 | 38.1 | 45.9 | 56.5 | 43.6 | 36.4 | 100.0 | 41.6 | 50.2 |
| L0H29 | −19.2 | −1.1 | −17.4 | −31.1 | −16.4 | −10.0 | −2.9 | −7.9 | −8.1 | −8.1 | −3.7 | +15.9 | −0.9 | −8.5 |
| L0H31 | −17.9 | −14.9 | −32.2 | −12.8 | −6.4 | −4.1 | −6.1 | +0.4 | +0.6 | −0.7 | +0.3 | −32.7 | −0.0 | −9.7 |
| L1H29 | −31.2 | −51.5 | −32.2 | −31.1 | −31.2 | −26.2 | −15.2 | −21.2 | −31.1 | −16.6 | −11.7 | −17.7 | +0.9 | −24.3 |
| L1H31 | −31.8 | −51.7 | −32.2 | −31.1 | −31.4 | −27.4 | −14.4 | −21.8 | −31.9 | −17.4 | −10.7 | −5.3 | +0.9 | −23.6 |

*Table 13.* Per-dataset results for Llama models across knockout conditions. Each cell shows mean with standard deviation below in parentheses. Baseline rows show un-ablated model performance.

| MODEL | TASK | GSM | AR | MBPP | HE | SWEAR | RHYME | HS | BQ | ARC-C | MMLU |
|---|---|---|---|---|---|---|---|---|---|---|---|
| LLAMA 3.1 8B | BASELINE | 78.5 | 85.3 | 58.4 | 68.3 | 100.0 | 65.5 | 79.3 | 84.1 | 55.2 | 68.0 |
| | GSM | 37.8 (6.8) | 25.1 (4.3) | – | – | – | – | 78.5 (0.6) | 83.3 (0.7) | 53.7 (2.4) | 67.5 (0.1) |
| | CODING | – | – | 46.7 (3.8) | 51.6 (3.4) | – | – | 78.6 (0.1) | 83.8 (0.1) | 54.1 (2.2) | 67.3 (0.1) |
| | SWEARING | – | – | – | – | 20.0 (6.0) | – | 78.5 (0.3) | 84.1 (0.3) | 55.0 (0.5) | 66.9 (0.3) |
| | RHYMING | – | – | – | – | – | 47.2 (4.5) | 76.2 (0.2) | 82.9 (0.3) | 51.3 (0.5) | 65.6 (0.6) |
| LLAMA 3.2 3B | BASELINE | 65.7 | 68.3 | 46.2 | 51.2 | 100.0 | 67.3 | 70.5 | 78.4 | 46.4 | 60.4 |
| | GSM | 24.8 (6.5) | 14.7 (0.4) | – | – | – | – | 70.2 (0.4) | 76.0 (2.3) | 44.8 (1.2) | 59.2 (1.0) |
| | CODING | – | – | 35.8 (5.9) | 47.2 (2.1) | – | – | 70.0 (0.7) | 77.1 (2.5) | 45.0 (0.4) | 59.4 (0.4) |
| | SWEARING | – | – | – | – | 64.0 (16.0) | – | 70.0 (0.5) | 76.6 (1.6) | 45.3 (1.2) | 59.1 (1.2) |
| | RHYMING | – | – | – | – | – | 47.2 (16.3) | 70.4 (0.3) | 78.3 (0.4) | 46.3 (0.6) | 59.7 (0.6) |
| LLAMA 3.2 1B | BASELINE | 32.9 | 51.7 | 32.2 | 31.1 | 100.0 | 41.6 | 60.7 | 69.5 | 38.1 | 45.9 |
| | GSM | 10.1 (6.7) | 49.7 (3.4) | – | – | – | – | 60.5 (0.1) | 67.9 (0.5) | 37.2 (0.5) | 44.5 (0.1) |
| | CODING | – | – | 25.3 (1.1) | 29.1 (2.3) | – | – | 59.4 (0.9) | 68.4 (0.7) | 36.4 (0.5) | 45.3 (0.1) |
| | SWEARING | – | – | – | – | 27.0 (25) | – | 60.0 (0.7) | 68.6 (0.1) | 37.7 (1.3) | 41.8 (1.8) |
| | RHYMING | – | – | – | – | – | 8.9 (7.7) | 60.4 (0.4) | 69.0 (1.0) | 37.9 (0.6) | 44.9 (0.9) |

*Table 14.* Per-dataset results for Qwen models across knockout conditions. Each cell shows mean with std below in parentheses. Baseline rows show un-ablated model performance.

| MODEL | TASK | GSM | MBPP | SWEAR | RHYME | HS | BQ | ARC-C | MMLU |
|---|---|---|---|---|---|---|---|---|---|
| | BASELINE | 63.5 | 53.6 | 100.0 | 29.2 | 74.9 | 80.0 | 48.0 | 65.5 |
| | GSM | 31.9 (9.4) | – | – | – | 74.8 (0.2) | 80.0 (0.7) | 47.1 (0.4) | 64.2 (0.3) |
| QWEN 3B | CODING | – | 4.6 (3.6) | – | – | 73.5 (1.4) | 78.7 (0.4) | 46.3 (1.1) | 62.7 (0.8) |
| | SWEARING | – | – | 78.2 (7.2) | – | 74.5 (0.5) | 79.7 (1.1) | 48.5 (0.9) | 65.2 (0.1) |
| | RHYMING | – | – | – | 11.8 (1.4) | 74.4 (0.2) | 79.3 (0.4) | 47.5 (0.8) | 64.8 (0.5) |
| | BASELINE | 82.6 | 37.6 | 100.0 | 38.9 | 80.4 | 86.3 | 54.6 | 71.7 |
| | GSM | 22.1 (4.4) | – | – | – | 78.6 (1.1) | 85.5 (0.8) | 54.2 (0.9) | 70.6 (0.3) |
| QWEN 7B | CODING | – | 2.7 (4.1) | – | – | 71.3 (6.2) | 82.4 (2.7) | 53.6 (1.2) | 71.1 (0.7) |
| | SWEARING | – | – | 38.8 (24.2) | – | 80.2 (0.9) | 86.2 (0.5) | 54.4 (0.4) | 71.7 (0.0) |
| | RHYMING | – | – | – | 11.2 (10.4) | 70.5 (7.0) | 84.2 (2.0) | 53.5 (0.6) | 70.4 (0.3) |
| | BASELINE | 45.8 | 36.0 | 100.0 | 56.6 | 84.5 | 88.1 | 62.7 | 78.8 |
| | GSM | 22.5 (4.5) | – | – | – | 84.2 (0.1) | 87.8 (0.2) | 62.4 (0.3) | 78.7 (0.2) |
| QWEN 14B | CODING | – | 0.0 (0.0) | – | – | 82.1 (0.1) | 87.6 (0.1) | 60.2 (0.3) | 75.6 (0.8) |
| | SWEARING | – | – | 67.4 (24.5) | – | 83.0 (1.3) | 87.6 (0.4) | 61.1 (1.3) | 76.9 (1.6) |
| | RHYMING | – | – | – | 26.5 (10.6) | 81.4 (1.1) | 87.4 (0.3) | 59.3 (1.7) | 75.8 (0.6) |

*Table 15.* Single-head performance across tasks. Knowledge-based multiple-choice tasks (Knowledge MC) are grouped separately from non-multiple-choice tasks.

| | KNOWLEDGE MC | | | | | NOT KNOWLEDGE MC | | | | | | | | | |
|---|---|---|---|---|---|---|---|---|---|---|---|---|---|---|---|
| HEAD | MMLU | W-B | W-CH | W-CY | AVG | GSM | AR | MBPP | HE | HS | BOOLQ | ARC C | SWEAR | RHYME | AVG |
| | | | | | | LLAMA-3.2-3B | | | | | | | | | |
| BL | 60.4 | 64.5 | 45.1 | 40.7 | 52.7 | 65.7 | 68.3 | 46.2 | 51.2 | 70.5 | 78.4 | 46.4 | 100.0 | 67.3 | 66.0 |
| L0H14 | −36.7 | −41.1 | −20.3 | −15.1 | −28.3 | −1.3 | +2.2 | −0.8 | −0.6 | −1.5 | −5.4 | +0.4 | +39.8 | −0.9 | +3.5 |
| L0H16 | −35.9 | −40.1 | −18.9 | −15.1 | −27.5 | +0.3 | −0.2 | −2.2 | −1.2 | +0.1 | −1.8 | +0.4 | +20.0 | 0.0 | +1.7 |
| | | | | | | LLAMA-3.2-1B | | | | | | | | | |
| BL | 45.9 | 56.5 | 43.6 | 36.4 | 45.6 | 32.9 | 51.7 | 32.2 | 31.1 | 60.7 | 69.5 | 38.1 | 100.0 | 41.6 | 50.9 |
| L0H22 | −21.3 | −30.5 | −17.6 | −14.0 | −20.8 | +1.1 | −0.6 | −0.8 | −1.2 | +0.3 | −1.5 | −0.1 | −14.8 | +1.8 | −1.8 |

*Table 16.* Task-specific heads identified in Llama 3.1 8B on various tasks, using various methods.

| TASK | METHOD | TOP 5 HEADS |
|------|--------|-------------|
| | | **MATH** |
| GSM8K | GREEDY | L16H21, L15H13, L18H18, L18H31, L0H28 |
| GSM8K | 1S-GREEDY | L16H21, L15H13, L13H18, L1H30, L0H28 |
| GSM8K | CS$_B$ | L15H13, L16H21, L13H18, L30H3, L9H19 |
| GSM8K | CS$_S$ | L15H13, L16H2, L12H12, L16H21, L13H18 |
| ARITH | GREEDY | L15H13, L16H21, L18H18, L7H27, L31H14 |
| ARITH | 1S-GREEDY | L15H13, L16H21, L14H0, L31H14, L13H7 |
| ARITH | CS$_S$ | L15H13, L16H21, L11H16, L13H16, L21H27 |
| | | **CODE** |
| MBPP | GREEDY | L24H31, L18H7, L7H6, L26H16, L24H14 |
| MBPP | 1S-GREEDY | L24H31, L6H19, L1H28, L12H30, L4H11 |
| MBPP | CS$_B$ | L4H13, L10H17, L24H31, L30H6, L8H16 |
| MBPP | CS$_S$ | L15H24, L1H28, L24H31, L31H24, L31H25 |
| | | **LANGUAGE** |
| SWEAR | GREEDY | L11H2, L1H17, L1H5, L0H13, L19H21 |
| SWEAR | 1S-GREEDY | L11H2, L9H10, L5H17, L10H18, L14H14 |
| SWEAR | CS$_B$ | L11H2, L23H24, L11H18, L11H15, L31H22 |
| SWEAR | CS$_S$ | L11H2, L26H7, L14H12, L1H1, L8H15 |
| RHYME | GREEDY | L0H29, L18H4, L0H28, L18H5, L13H28 |
| RHYME | 1S-GREEDY | L0H29, L18H5, L28H25, L16H19, L10H22 |
| RHYME | CS$_B$ | L0H29, L29H27, L28H8, L29H0, L30H15 |
| RHYME | CS$_S$ | L0H29, L9H10, L20H1, L14H12, L23H4 |
| | | **WMDP** |
| BIO | GREEDY | L3H22, L10H3, L11H13, L31H14, L1H5 |
| BIO | 1S-GREEDY | L4H16, L3H22, L31H14, L1H24, L6H24 |
| BIO | CS$_B$ | L13H5, L25H18, L31H14, L12H6, L23H13 |
| BIO | CS$_S$ | L10H26, L12H22, L9H9, L21H26, L29H2 |
| CHEM | GREEDY | L0H3, L8H8, L5H3, L10H23, L9H21 |
| CHEM | 1S-GREEDY | L0H3, L7H7, L12H4, L2H22, L16H22 |
| CHEM | CS$_B$ | L19H31, L13H18, L9H29, L22H23, L16H22 |
| CHEM | CS$_S$ | L0H24, L0H3, L20H14, L7H21, L18H30 |
| CYBER | GREEDY | L31H14, L11H20, L13H2, L8H9, L7H31 |
| CYBER | 1S-GREEDY | L31H14, L13H16, L10H29, L5H19, L7H16 |
| CYBER | CS$_B$ | L11H14, L7H31, L13H16, L15H7, L8H23 |
| CYBER | CS$_S$ | L31H14, L24H14, L27H6, L11H0, L27H13 |

*Table 17.* Task-specific heads identified across various tasks and models. All heads are found using stratified compressed sensing.

| TASK | HEADS (TOP 5) |
|------|---------------|
| **LLAMA 3.2 3B** | |
| GSM8K | L16H22, L14H10, L8H8, L12H1, L18H5 |
| ARITHMETIC | L16H22, L14H10, L8H15, L11H19, L25H12 |
| MBPP | L4H6, L17H11, L1H16, L3H20, L0H16 |
| SWEARING | L22H12, L0H2, L1H6, L5H0, L9H11 |
| RHYMING | L16H3, L18H19, L0H0, L0H4, L18H2 |
| WMDP BIO | L10H26, L12H22, L9H9, L21H26, L29H2 |
| WMDP CHEM | L0H24, L0H3, L20H14, L7H21, L18H30 |
| WMDP CYBER | L31H14, L24H14, L27H6, L11H0, L27H13 |
| **LLAMA 3.2 1B** | |
| GSM8K | L15H30, L8H30, L5H12, L6H5, L5H3 |
| ARITHMETIC | L11H15, L11H13, L3H6, L11H14, L9H23 |
| MBPP | L3H25, L8H31, L2H27, L10H4, L8H12 |
| SWEARING | L7H14, L15H14, L6H29, L0H2, L9H1 |
| RHYMING | L5H7, L9H2, L9H27, L5H19, L4H31 |
| WMDP BIO | L0H22, L11H8, L2H2, L15H22, L9H26 |
| WMDP CHEM | L0H22, L9H26, L0H4, L11H25, L10H26 |
| WMDP CYBER | L0H22, L6H31, L6H16, L7H28, L5H22 |
| **QWEN 2.5 3B** | |
| GSM8K | L27H6, L27H0, L27H1, L18H9, L20H10 |
| MBPP | L0H2, L8H11, L33H11, L0H12, L23H4 |
| SWEARING | L21H10, L17H13, L17H11, L13H0, L30H6 |
| RHYMING | L32H7, L28H11, L4H5, L15H15, L26H9 |
| **QWEN 2.5 7B** | |
| GSM8K | L0H0, L0H15, L20H6, L0H25, L1H20 |
| MBPP | L17H13, L19H0, L8H17, L0H23, L11H0 |
| SWEARING | L18H17, L16H8, L7H26, L15H8, L27H14 |
| RHYMING | L0H24, L0H25, L14H16, L25H8, L0H1 |

