# OpenReview forum: "Compressed Sensing for Capability Localization in Large Language Models"
_ICML.cc/2026/Conference — ICML 2026 regular_

### Official Review · Reviewer_5xXy · 2026-02-14

**Soundness:** 3
**Presentation:** 3
**Significance:** 3
**Originality:** 3
**Overall Recommendation:** 4
**Confidence:** 4

**Summary:**

The paper studies where high-level capabilities reside within transformer-based LLMs and argues that many are highly localized to sparse sets of attention heads. It introduces a compressed sensing approach that identifies these heads by evaluating the model under randomized multi-head ablations and solving a sparse regression (Lasso), enabling up to 50× fewer evaluations than greedy search. Empirically, knocking out as few as five identified heads can reduce performance on targeted tasks (math, code, swearing, rhyming) by large margins while mostly preserving unrelated abilities; the paper also reports “universal heads” that broadly affect many tasks and scale-dependent localization phenomena on WMDP.

**Compliance With Llm Reviewing Policy:**

Affirmed.

**Final Justification:**

I tend to retain my original opinions and scores

**Key Questions For Authors:**

1. How is the Lasso regularization strength λ chosen? Is β0 fixed to the baseline accuracy or fit jointly? Please report sensitivity to λ and the number/density of masks.

2. Can you provide random-head ablation baselines (mean ± std over multiple random sets) for each task and model, and per-benchmark breakdowns (not just averages) of collateral effects?

3. How stable are the identified heads across multiple seeds, dataset subsamples (beyond 100 examples), prompt templates, and decoding settings? Please add confidence intervals.

4. Please explain the difference between you and [1], as well as your strengths.

[1] Capability Localization: Capabilities Can be Localized rather than Individual Knowledge.

**Limitations:**

The authors did not discuss limitations and potential social impacts, please add it.

**Strengths And Weaknesses:**

Strengths:

1. The formulation of head-importance recovery as a compressed sensing problem with randomized ablation masks and L1-regularized regression is novel and elegant for capability localization.

2. Demonstrates consistent task-specific degradation with small numbers of head knockouts across multiple model families (Llama, Qwen) and scales (1B–8B).

3. Problem setup, assumptions, and algorithm are presented clearly; the linear model, mask construction, and recovery criterion are easy to follow.

Weaknesses:

1. The linear additivity assumption and use of dataset-level accuracy as a noisy, non-linear response variable are not deeply stress-tested; theoretical guarantees (RIP/mutual coherence) for the proposed measurement designs are not provided.

2. Custom “swearing” and “rhyming” datasets seem very small (e.g., 12 prompts) and possibly brittle; no confidence intervals or multiple-seed evaluations are reported, making large percentage drops difficult to interpret statistically.

3. Minor inconsistencies/typos (e.g., duplicated model names, “IS/1S-Greedy” naming) and occasional ambiguity in methodological details (e.g., whether the intercept is fixed or fit, exact sparsity per mask).

---

> ### Author Rebuttal · Authors · 2026-03-31
>
> We thank the reviewer for their thoughtful comments and positive review. We are glad that they recognized our approach as novel and elegant and that they found the presentation easy to follow. We address the weaknesses and questions below:
>
> ## Linear additivity assumption [W1]
>
> We agree that theoretical guarantees would add depth but we leave this to future work. We agree that the linear additivity assumption may seem tenuous, but we believe it is sufficient for our use case. Notably, most of the performance degradation across tasks is caused by ablating only one or two attention heads. We do not claim (and will emphasize in the final paper) that all heads can be combined linearly to determine influence. We only claim that we can approximate the contribution of task-specific heads as linear. We agree that further investigation into this assumption would be useful.
>
> A related method, ProxySPEX [1], does not assume linearity and models higher-order interactions, but requires at least 50x more evaluations. The higher-order interactions are useful in this setting since the goal of the method is to reconstruct the entire output signal. Since our aim is only to find a sparse set of task-specific heads, the higher-order information may be omitted for efficiency.
>
> ## Statistical significance and head stability [W2, Q3]
>
> We report average $\pm$ standard error (SE) over four random seeds of dataset subsamples. Task-specific degradation is statistically significant in all settings. Qwen results will be added to the final paper.
>
> | Model | Task | Δ Task | Δ Gen |
> |---|---|---|--|
> | Llama-3.1-8B | GSM8k | -34.14 ± 2.34 | -0.95 ± 0.37 |
> | Llama-3.2-3B | GSM8k | -40.95 ± 2.68 | -1.57 ± 0.55 |
> | Llama-3.2-1B | GSM8k | -22.83 ± 2.83 | -1.51 ± 0.49 |
> | Llama-3.1-8B | MBPP | -9.65 ± 2.55 | -0.64 ± 0.46 |
> | Llama-3.2-3B | MBPP | -10.55 ± 2.41 | -0.95 ± 0.32 |
> | Llama-3.2-1B | MBPP | -13.25 ± 6.34 | -2.48 ± 1.32 |
> | Llama-3.1-8B | Swearing | -81.00 ± 3.00 | -0.51 ± 0.09 |
> | Llama-3.2-3B | Swearing | -29.00 ± 9.50 | -1.02 ± 0.48 |
> | Llama-3.2-1B | Swearing | -79.00 ± 12.00 | -1.61 ± 0.20 |
> | Llama-3.1-8B | Rhyming | -22.35 ± 4.46 | -2.71 ± 0.12 |
> | Llama-3.2-3B | Rhyming | -16.37 ± 7.62 | -0.60 ± 0.38 |
> | Llama-3.2-1B | Rhyming | -34.95 ± 3.85 | -0.74 ± 0.23 |
>
> The most important 1-2 heads are consistently recovered across seeds, while lower ranked heads vary. As we increase the number of masks, head stability improves. We will discuss this as a tunable hyperparameter in the final paper.
>
> We assess stability in Llama-3.1-8B by checking whether the heads most consistently identified by greedy search (Table 9) are recovered across seeds of our compressed sensing method. For GSM8K, across four seeds, at least one of the top two heads are present in each set of ranked top 5 heads. For MBPP, the top head is found in two out of the four sets. For swearing, the top head is ranked first for all four seeds. In rhyming, the top head is present in all four sets and ranked first in three.
>
> ## Hyperparameters [W3, Q1]
>
> For Lasso, the intercept is fit jointly. The regularization strength is 0.01. Higher regularization results in more (or all) coefficients being set to zero and therefore rendering our ranking approach useless.
>
> Number of masks / sparsity for each model and dataset:
> - Llama 3.1 8B: GSM8k: 100/0.02, MBPP: 200/0.01, Swearing: 200/0.01, Rhyming: 100/0.05.
> - Llama 3.2 3B: GSM8k: 200/0.01, MBPP: 200/0.01, Swearing: 100/0.02, Rhyming: 200/0.02.
> - Llama 3.1 1B: GSM8k: 200/0.01, MBPP: 200/0.01, Swearing: 200/0.02, Rhyming: 100/0.02.
> - Qwen 7B: GSM8k: 100/0.02, MBPP: 100/0.02, Swearing: 200/0.01, Rhyming: 100/0.02.
> - Qwen 3B: GSM8k: 200/0.02, MBPP: 200/0.05, Swearing: 200/0.05, Rhyming: 100/0.02.
>
> We will clear up the minor typos and inconsistencies.
>
> ## Random ablation and benchmark breakdowns [Q2]
>
> Ablating five random heads has minimal impact across tasks.
>
> | Task | % Drop (Mean ± Std) |
> |---|---|
> | GSM8K | -5.99 ± 5.82 |
> | Arithmetic | 0.93 ± 3.90 |
> | MBPP | -1.15 ± 2.28 |
> | HumanEval | -2.68 ± 1.79 |
> | HellaSwag | -0.78 ± 0.53 |
> | BoolQ | -0.53 ± 0.13 |
> | ARC-C | -2.77 ± 3.49 |
> | MMLU | -0.21 ± 0.37 |
>
> We will add per-benchmark performance in the final paper.
>
> ## Capability localization [Q4]
>
> [2] presents an interesting method that is complementary to our work. They find that capabilities are localizable while individual facts are not. While their method operates at the MLP neuron level, our paper operates at a coarser granularity. The strengths of our work are the direct causal evidence via ablation, an efficient gradient-free method, and broad scope of models and tasks.
>
> [1] Butler L, Agarwal A, Kang JS, Erginbas YE, Yu B, Ramchandran K. Proxyspex: Inference-efficient interpretability via sparse feature interactions in llms. NeurIPS 2025.
>
> [2] Huang, X., Liu, J., Wang, Y., Zhao, J., & Liu, K. (2025). Capability Localization: Capabilities Can be Localized rather than Individual Knowledge. arXiv preprint arXiv:2502.20992.

---

> > ### Author Rebuttal · Reviewer_5xXy · 2026-04-03
> >
> > The authors have addressed my concerns. However, based on the overall innovation and feedback of the paper, I tend to maintain the original score.

---

> > > ### Author Response · Authors · 2026-04-08
> > >
> > > We would like to thank the reviewer for taking the time to review our rebuttal and confirming that their concerns were resolved. We appreciate the feedback and support of our work.

---

### Official Review · Reviewer_dA6f · 2026-02-26

**Soundness:** 1
**Presentation:** 2
**Significance:** 3
**Originality:** 3
**Overall Recommendation:** 5
**Confidence:** 3

**Summary:**

The paper introduces an efficient compressed-sensing–based method to identify "task-specific heads" in Large Language Models, demonstrating that high-level capabilities are highly localized. By performing strategic "knockouts" (ablations) of random subsets of attention heads and measuring the resulting performance drops, the authors can pinpoint a tiny number of heads—sometimes as few as one to five—that are essential for specific tasks like arithmetic, coding, or rhyming. The results show that ablating these few heads can degrade performance on targeted benchmarks by up to 65% while leaving unrelated capabilities intact, suggesting that LLM architectures naturally organize into sparse, modular functional units.

**Compliance With Llm Reviewing Policy:**

Affirmed.

**Final Justification:**

The authors provide fullsome feedback on my questions. They commit to "clarify the paper to avoid misrepresentation of our (MI) claims". On this basis, I am revising my score to reflect the main thrust of the paper which is the novel, efficient, useful method presented.

**Key Questions For Authors:**

I wish I could approve this paper but as it stands I can't. Please keep improving it.

Consider looking inside one the task-specific heads to see what it is doing.

Specifically, consider using the techniques from this paper http://arxiv.org/abs/2511.20273 "Vector-Based Interpretability of Transformer Circuits" which says "Nodes in a computational graph, that are previously identified as circuit elements show strong activation along specific low-rank directions, suggesting that meaningful computations reside in compact subspaces." It matches single vectors in single heads with task use cases.

**Limitations:**

Yes

**Strengths And Weaknesses:**

Strengths:
- “A key contribution of our work is the development of efficient algorithms for identifying task-specific heads”. I accept and appreciate the new technique and empirical results. This is great, novel and useful. I think you should make the technique public as people (including me) can use it!
- Size and number of models in experiments
- Number of benchmarks in experiments.
- Table 2 showing that ablating 1 “maths” head reduces Arithmetic benchmark performance by 65%, reduces GSM8K by 25%, with other benchmarks mostly unaffected

Weaknesses:
- I think many of your Mech Interp conclusions are unsupported. A root cause of this is you have assumed that because a head is “required” to complete a task that it is also “sufficient” to complete that task. This  doesn’t follow. Assuming “sufficiency” you claim “We show that many capabilities are highly localized to small subsets of attention heads”. I don’t accept that the circuits implicated in *calculating* 65% of the Arithmetic benchmark performance reside in one attention head.
- It is more likely that one of interesting attention heads you have discovered is not an “Calculation head” but a “Categorization head” containing a vector for each of “This is an addition prompt”, “This is a subtraction prompt”, “This is multiplication prompt”, etc - a control mechanism. These features are anticorrelated (at most *one* activates for each prompt) so they are suitable to be polysemantically combined into one head. Many Arithmetic use case categorization features could be packed into one attention head. Ablating that one attention head would impact all these features by breaking the control mechanism. As a “Categorization head”, the head is *required* for the calculation but not *sufficient*.
- The “universal heads” are interesting but the paper would be strengthened with a theory for what are they for. They are obviously “required” but what is their purpose? Needs more thought.
- The “Scale dependence of localization” seems empirical rather than an insight. Differences between models should be an expected starting point. Also, I expect different models to pack their Calculation heads in different ways just based on random training differences. Another potential factor: If a task space (say Arithmetic) has sufficient “task prompt use cases” to nearly fill the carrying capacity of an attention head, then I would hope the model would pack them into one head as their anticorrelation gives computational efficiencies. But if a task space (say Chem) does not have many task prompt use cases I wouldn’t be surprised if tasks from multiple uses cases were packed into one general-purpose “Categorization head” (for some but less computational efficiencies). Rethink the interpretations from this viewpoint.

---

> ### Author Rebuttal · Authors · 2026-03-31
>
> We would like to thank the reviewer for their thoughtful feedback. We are glad to hear that they found our work “great, novel, and useful” and could use it in their own work. We have code prepared and will release it with the final paper.
>
> We mostly agree with the reviewer’s comments and think that they offer promising directions for future work and interesting additional analyses. We hope that the clarifications below will address the reviewer’s core concerns and warrant a revised score.
>
> ## Mechanistic interpretability claims [W1]
> We agree with the reviewer here and will clarify the paper to avoid misrepresentation of our claims. We claim that the heads are necessary for the given task but we certainly do not want to claim that the heads are sufficient. In fact, we ran preliminary experiments retaining only a few task-specific heads (and ablating most heads in the network) and the resulting performance was quite poor. We will add clarification in the paper, such as by framing capability localization as the localization of some task-critical behaviors rather than suggesting that all of the processing for the given task occurs in the identified heads. Our task-specific heads do not make up a circuit, but rather are sparse subsets of the model that are critical for strong task-specific performance.
>
>
> ## Alternate head interpretations [W2]
> Again, we agree with the reviewer that this is a possible interpretation. Our work aims to identify heads where capabilities are localized and we hope that our work can be used as part of the pipeline for more in-depth mechanistic studies of attention heads. We believe that our contribution of an efficient localization method is a useful tool for the interpretability community, independent of any specific mechanistic analysis.
>
> Some evidence that the identified head is at least arithmetic-specific is shown in Table 3 in our paper, where heads identified using GSM8k also impact performance on the Arithmetic dataset, and vice versa. We would also like to point the reviewer toward a relevant paper that identifies similar arithmetic heads to our method and analyzes the attention patterns to determine the specific behavior implemented by each head [1]. In Appendix B.2 ``... [E]ach of the three heads attends to a single input token, copying the representation from that position and projecting it to the last position. Specifically, L16H21 attends to the first operand, L2H2 attends to the operator and L15H13 attends to the second operand. This implies the role of each such head is to move the representation from that position, which includes to the last position, where it is further processed by the bag of heuristics implemented in the middle- and late-layer MLPs. ‘’ Our work identifies L16H21 and L15H13 as the top arithmetic-specific heads. We hope that our work can be used to further facilitate deeper analysis, including mechanistic understanding.
>
> ## Universal heads [W3]
> The main contribution of our paper is an efficient method for localizing task-specific heads and a preliminary discussion of some observed phenomena. However, we agree that the analysis of the universal heads could be more complete and we will add additional analysis to the final paper. As a starting point, we run a small set of experiments to analyze attention patterns across datasets for the universal heads in Llama-3.1-8B and find evidence that L0H31 overwhelmingly attends to the BOS token. L0H31 may be playing a role in concentrating attention towards the start of the sequence to facilitate attention sink patterns. While deeper analysis is necessary to make concrete claims, we aim to demonstrate how our methods and findings can be used as part of a mechanistic interpretability pipeline.
>
>
> ## Scale dependence and interpretability of task-specific heads [W4, Q1]
> The reviewer’s hypothesis that anticorrelated features may be packed into shared heads at smaller scales is consistent with our finding of knowledge-based multiple-choice heads in the smaller models, where a single head mediates performance across WMDP and MMLU simultaneously. We view the empirical documentation of this scale-dependent pattern as a contribution in itself, and the reviewer’s proposed mechanistic explanation offers a promising direction for further study. We agree that applying sub-head decomposition methods such as [2] to the heads identified by our method is a natural next step and could inform whether the task-specific heads we identify implement dedicated computations or serve as routing mechanisms, as the reviewer suggests.
>
>
> [1] Nikankin, Y., Reusch, A., Mueller, A., & Belinkov, Y. (2024). Arithmetic without algorithms: Language models solve math with a bag of heuristics. arXiv preprint arXiv:2410.21272.
>
> [2] Ahmad, A., Joshi, A., & Modi, A. (2025). Beyond Components: Singular Vector-Based Interpretability of Transformer Circuits. arXiv preprint arXiv:2511.20273.

---

> > ### Author Rebuttal · Reviewer_dA6f · 2026-04-02
> >
> > I thank the authors for their fullsome feedback. Assuming the committment to "clarify the paper to avoid misrepresentation of our claims" has been implemented (including the abstract which says "We show that many capabilities are highly localized to small subsets of attention heads within Transformer architectures" and would better read "highly dependent on"), I've revised my score to reflect the most significant aspect of this paper - the novel, efficient, useful new method.

---

> > > ### Author Response · Authors · 2026-04-08
> > >
> > > We would like to thank the reviewer for their productive discussion of our work. We are glad that they found our rebuttal thorough and valuable and chose to raise their score. We appreciate the constructive feedback and support of our work.

---

### Official Review · Reviewer_9SoK · 2026-03-10

**Soundness:** 3
**Presentation:** 3
**Significance:** 3
**Originality:** 3
**Overall Recommendation:** 5
**Confidence:** 4

**Summary:**

The paper proposes an efficient method to identify important attention heads for specific tasks motivated by compressed sensing. By randomly ablating subsets of attention heads, a relatively small number of model evaluations is sufficient to identify heads that—once removed—would tank task performance.

The method is applied on Llamas and Qwen of 1B to 8B scale. It is discovered that removing only 5 heads of hundreds can drastically lower task performance but keep general capability almost the same. They also discover “universal” heads that cannot be removed without severely harming capabilities across the board. Results indicate that sparsity of heads is influenced by scale of the models, with larger models having more specialized heads. Further, grouped query attention may influence where task-specific heads form.

**Compliance With Llm Reviewing Policy:**

Affirmed.

**Final Justification:**

During the rebuttal, a few important points were brought up:
1. The claim of "localization" is not fully supported. This was addressed in the response to dA6f and the claim must be adequately weakened.
2. The methodological novelty is indeed reduced by the ProxySPEX paper. However, the LASSO method was not discussed in detail nor its tradeoffs compared to ProxySPEX. Hence, I think it is still valuable for the community to lay out the method in more detail and highlight the efficiency of it.
3. The instability mentioned as the first point in the justification of GHP3 should indeed be mentioned and it would be good if the authors do a more detailed analysis of the stability of found heads (e.g., not only measuring recall@k with the greedy method as gold standard, but also check how often the top 2 heads are in the top 10 identified heads by the efficient method).

I have reduced my presentation and soundness scores to 3. Given the rebuttals, I think the final paper will meet the quality standard.

**Key Questions For Authors:**

**Q1:** Could you elaborate on scaling this method to larger models? What technical constraints are there? What is holding you back? Since you propose a general method, it would be really cool to have a “recipe” for applying the method. Please include hyperparameter tuning steps etc. Having this would increase the chance of adoption in the community.

**Q2:** What are your plans for releasing code for the method? Especially for contributions around new methods, releasing a working code example is important.

**Limitations:**

yes

**Strengths And Weaknesses:**

## Strengths

-	Excellent presentation. I found the paper very easy to read.
-	Convincing evidence for the claims of sparsity.
-	Extensive related work.
-	Method appears to be much easier to use than circuit analysis (but see questions).
-	Skill localization is important: it would be cool to see localization of safety-related/misalignment skills. It seems practically feasible to use this method for many more experiments, such as regarding when and where does sparsity emerge first. It could be quite useful for further research.

## Weaknesses

-	No code. (If you uploaded it to openreview, it is not accessible to me as a reviewer.)
-	Experiments are relatively small in scale. Since one claim of the paper is that the method is efficient, I would expect results with a larger model as well. I see how this is a typical reviewer complaint to always make experiments bigger and more costly, but it would be really interesting and lend credence to the claim of efficiency (see questions).
-	Evaluation: I would like to see overlap statistics between heads identified with your efficient method vs greedy. Are there any patterns regarding finding heads at all or ordering in terms of importance? This would give some more information about what the tradeoff between the different approaches is.
-	Evaluation: what is the effect of zeroing the output vs using the average value?

---

> ### Author Rebuttal · Authors · 2026-03-31
>
> We would like to thank the reviewer for their comprehensive comments and positive recommendation. We are glad that they find the work important and backed up by strong evidence, and we appreciate that they found the paper easy to read. The reviewer’s comments regarding safety-related skills are a promising direction of future work and our WMDP experiments in Section 6.2 are a step in that direction. We address the main weaknesses/questions below.
>
> ## Releasing code [W1, Q2]
> We will upload our code. We have it prepared and will release it with the final version of the paper.
>
> ## Scaling and method ``recipe’’  [W2, Q1]
> We were unable to run larger models in this time frame; the main constraint we are facing is limited compute resources. We believe that the trends we identified will hold at a larger scale and will run a larger model for the final version of the paper.
>
> We will add more in-depth hyperparameter tuning details and method guidelines by adding the following to our paper:
> - Step 1: Define the evaluation function. Choose a task-specific dataset and evaluation metric (eg. accuracy on LMEval tasks such as GSM8k and MBPP, or number of swear words generated for our custom swearing task). For the preexisting LMEval tasks, we found 100 examples to be a reasonable data subset but we did not perform extensive hyperparameter tuning. For swearing and rhyming, we randomly sample approximately half of the prompts (6/12 for swearing and 50/113 for rhyming).
> - Step 2: Set up hyperparameter search on the stratified sampling algorithm. A reasonable lower bound on masks and sparsity is 100 masks, each with 0.01 sparsity, since this allows each head to be measured once. The number of masks should be as small as possible while providing enough measurements for head identification. Sparsity should be relatively low as large sparsities can degrade overall network performance and obscure individual head contributions. Recommended initial values are masks = [100, 200, 300] and sparsity = [0.01, 0.02, 0.05].
> - Step 3: Run identification on each hyperparameter combination. Construct the stratified measurement matrix, evaluate the model (on the specified data subset from Step 1) under each mask configuration, and solve the Lasso regression. We use Lasso from scikit-learn with alpha=0.01, max_iter=5000, and other settings at their default values (notably, fit_intercept=True). Select the top k heads with the largest negative coefficients.
> - Step 4: Validate and adjust if needed. Ablate the identified top-k heads (in our case, we set k=5) and evaluate on the same data subset. Compare results across number of masks and sparsity to determine the best hyperparameters for the task.
> - Step 5: Final evaluation. Once hyperparameters are fixed, evaluate the ablated model on the full dataset to confirm the effect and measure collateral impact on unrelated benchmarks.
>
> ## Overlap between efficient and greedy heads [W3]
>
> We have listed all heads in Table 9 in the Appendix and summarize the takeaways here: The same top two GSM8k heads (L16H21 and L15H13) are identified as #1 or #2 rank by two greedy variants and one efficient variant (the second efficient variant places L16H21 at rank 4). Beyond that, L13H18 shows up in both efficient variants and one greedy variant. For MBPP, the head L24H31 is the top head identified for both greedy variants and is located at rank 3 for both efficient variants. For swearing, the same top head (L11H2) is identified by all four method variants. For rhyming, the same top head (L0H29) is also identified by all four method variants. Because we select the top five heads rather than just the top 1, our method is robust to slight reordering. The most important heads are reliably recovered within the top five even when the exact rank differs across methods.
>
> ## Zero ablation vs mean ablation [W4]
>
> Zero ablation and mean ablation of top 5 GSM8k heads in Llama-3.1-8B have similar results. Average degradation $\pm$ standard deviation is reported across four random seeds of dataset subsample.
>
> | Ablation | GSM8K | Arithmetic | Gen |
> |----------|--------|------------|------|
> | Zero | -43.5 ± 4.67 | -54.1 ± 9.26 | -1.3 ± 0.73 |
> | Mean | -36.3 ± 5.86 | -55.4 ± 9.41 | -1.03 ± 0.48 |

---

> > ### Author Rebuttal · Reviewer_9SoK · 2026-04-03
> >
> > All my concerns were addressed. Not running the experiments with a larger model is understandable given the time frame and I trust the author's promise to include it in a final version.

---

> > > ### Author Response · Authors · 2026-04-08
> > >
> > > We would like to thank the reviewer for taking the time to review our rebuttal and confirming that their concerns were resolved. We appreciate the feedback and will be sure to include larger scale experiments in a final version.

---

### Official Review · Reviewer_GPH3 · 2026-03-11

**Soundness:** 3
**Presentation:** 3
**Significance:** 2
**Originality:** 1
**Overall Recommendation:** 3
**Confidence:** 5

**Summary:**

This paper investigates the problem of "head selection" for attention-based large language models. The authors identify important heads via a compressed sensing approach and compare against a naive greedy non-sparse baseline. The authors find taks helpful for knowledge-based multiple choice and mathematics. The authors also identify a set of

**Compliance With Llm Reviewing Policy:**

Affirmed.

**Final Justification:**

I have decided to keep my score below the acceptance threshold after some final thoughts and looking at the final rebuttal reply that was posted today. The promised changes and experiments definitely are a step in the right direction for this manuscript, and I think expose some important points. However, I have some serious concerns after seeing them.

1. Looking at the recall scores were quite surprising. If accepted, **I must insist that the authors include these recall results**. It exposes a critical weakness: the algorithm is quite unstable with such few evaluations. Since the authors interpret these top heads as being "important for the task" it should be the case that the algorithm outputs the same heads with some reasonable reliability.

2. I realized after looking at the recall results, that the authors task of removing heads to *decrease accuracy* is actually much simpler than removing heads to *maintain accuracy* (this is the experiment that is done in ProxySPEX). My intuition tells me that trying to "maintain accuracy" is the superior experiment (there are many ways to get the model to perform worse, but it is probably much harder to remove many heads and keep the model performing well on a specific task). I would do some experiments about this if I had the time, but unfortunately, my reply was due less than 24hrs after I saw these results. The experiments of the authors provide some insights though.

One question I see: are there disjoint (or nearly disjoint) sets of heads that we can remove that can both significantly decrease the accuracy? With such low recall and low variance that we see in the new experiments, *it seem that this must be the case*. If so, what does this mean for the interpretation as "task specific heads" if I can run your method again, and get another, mostly distinct set of "task specific heads".

**Key Questions For Authors:**

1. What would happen if you compare with [1] above? It seems that the code is available online, and it could be interesting to see redundancy, etc. Does capturing interactions help you uncover anything interesting in your experiments?

2. For the LASSO based experiments, can you run for larger number of evals? With 100 evals, what is the stability of the top-k heads?

**Limitations:**

Discussion of limitations is pretty limited, and this work could do with much more. I think taking a deeper look into the sparsity and additivity assumption, the later of which is somewhat in contrast to prior works.

**Strengths And Weaknesses:**

## Strengths

1. I think this idea is great. It would be very interesting to better understand the phenomenon of localization in transformer models.
2. Compressed sensing seems like an ideal framework for reducing the cost of identifying important head, since sparsity is clearly present (at least in models of the vintage studied here).

## Weaknesses

1. This paper is part of a very well-studied and standard machine learning problem: *feature attribution* (the output of each head is a feature), but the authors do not acknowledge this. A proper discussion of this is mandatory and more relevant than the existing related work, which contains a lot of very tangentially related work (suggest you cut this down).
2. The author's claim that they "introduce a compressed sensing based method that exploits the sparsity of these heads to identify them via strategic knockouts and a small number of model evaluations". In [1] with a very similar experimental setup, the LASSO is used as a baseline approach for head identification.
3. In reference to [1] the authors don't explicitly make the assumption of *additivity*. I think it is pretty clear that additivity is not a good assumption to make based on the figures we see in [1], though it may be suitable for extracting only the top-5 heads.
4. The authors need to be scientific in their presentation of the results. Are the results in the graphs and tables significant? Authors should report confidence intervals / standard error.
5. I would be very excited by this work if it was more comprehensive. Can we hypothesis test fact localization (for different types of facts) in a rigorous way? (get similarity between datasets and compare with similarity of head attribution)
6. This paper currently reads as an exploratory study. It would be better if the authors could clearly state the main contributions and the evidence for these contributions.

[1] Butler L, Agarwal A, Kang JS, Erginbas YE, Yu B, Ramchandran K. Proxyspex: Inference-efficient interpretability via sparse feature interactions in llms. NeurIPS 2025.

---

> ### Author Rebuttal · Authors · 2026-03-31
>
> We thank the reviewer for their thoughtful comments. We are glad that they found the idea interesting and believe that compressed sensing is an ideal approach to the problem. We hope that the reviewer will find the following discussion satisfactory and consider revising their score.
>
> ## Comparison to ProxySPEX [W1, W2, W3, Q1]
>
> We thank the reviewer for pointing out the connection to feature attribution and will add a discussion of this literature, including ProxySPEX, to the revised manuscript.
>
> Our approach is similar in some ways to ProxySPEX but primarily differs in the goal of the method. ProxySPEX aims to reconstruct the entire output signal but our goal is only to identify a sparse set of task-specific heads. ProxySPEX could be used to identify task-specific heads, but is much more expensive. The attention head procedure outlined in Section 5.2 uses 5000 evaluations relative to our 100-200 evaluations. This expense is due to modeling higher order interactions, which are critical for reconstructing the output but unnecessary for identifying task-specific heads, which is why we can utilize the additivity assumption. Since we are only identifying a small handful of important heads, first order effects are good enough. The top one or two heads often contributes to most of the performance degradation upon ablation. Figure 2 in our paper shows that the majority of the task-specific performance degradation occurs after ablation of the top head. Given this behavior of the task-specific heads, additivity is a fine assumption in our case, and allows us to achieve significant efficiency gains over an approach like ProxySPEX.
>
> We run ProxySPEX with a budget of 100 evaluations to localize GSM8k heads in Llama-3.1-8b-it. We restrict search to layers 14-16, which includes the top two heads identified by our method (L16H21 and L15H13). The top heads identified by ProxySPEX in this setting are L16H18, L16H1, L14H11, L15H26, and L16H17. Neither of our top two heads are located anywhere in the top 20 heads identified by ProxySPEX. Ablating the top five ProxySPEX heads results in minimal impact on math and general-purpose tasks. Relative to baseline values, ablating these heads result in -0.84% change on GSM8k and +0.38% on Arithmetic. Although we chose 100 evaluations to match our computational budget, this was insufficient to identify the most important heads, as indicated by no overlap with our identified heads and by no downstream impact of ablating the top five heads. This demonstrates that at matched budgets, our stratified design recovers meaningful heads where ProxySPEX does not. Higher-order interactions may offer little to no benefit in our current setting.
>
> ## Confidence intervals [W4]
> We report average $\pm$ standard error (SE) over four random seeds of dataset subsamples. Task-specific degradation is statistically significant in all settings. Qwen results will be added to the final paper.
>
> | Model | Task | Δ Task | Δ Gen |
> |--------------|----------|-------------------|------------------|
> | Llama-3.1-8B | GSM8k | -34.14 ± 2.34 | -0.95 ± 0.37 |
> | Llama-3.2-3B | GSM8k | -40.95 ± 2.68 | -1.57 ± 0.55 |
> | Llama-3.2-1B | GSM8k | -22.83 ± 2.83 | -1.51 ± 0.49 |
> | Llama-3.1-8B | MBPP | -9.65 ± 2.55 | -0.64 ± 0.46 |
> | Llama-3.2-3B | MBPP | -10.55 ± 2.41 | -0.95 ± 0.32 |
> | Llama-3.2-1B | MBPP | -13.25 ± 6.34 | -2.48 ± 1.32 |
> | Llama-3.1-8B | Swearing | -81.00 ± 3.00 | -0.51 ± 0.09 |
> | Llama-3.2-3B | Swearing | -29.00 ± 9.50 | -1.02 ± 0.48 |
> | Llama-3.2-1B | Swearing | -79.00 ± 12.00 | -1.61 ± 0.20 |
> | Llama-3.1-8B | Rhyming | -22.35 ± 4.46 | -2.71 ± 0.12 |
> | Llama-3.2-3B | Rhyming | -16.37 ± 7.62 | -0.60 ± 0.38 |
> | Llama-3.2-1B | Rhyming | -34.95 ± 3.85 | -0.74 ± 0.23 |
>
> ## Contributions [W6]
>
> Please refer to Section 7. We will release code with the final paper.
>
> ## More evals and stability [Q2]
>
> Results for varying numbers of evals on Llama-3.1-8B GSM8k show a tradeoff between accuracy and efficiency.
>
> | # Evals | GSM8K | Arithmetic | Gen |
> |--------|--------|------------|------------|
> | 100 | -43.5 ± 4.67 | -54.1 ± 9.26 | -1.3 ± 0.73 |
> | 200 | -44.1 ± 10.34 | -66.3 ± 10.11 | -1.2 ± 0.42 |
> | 300 | -52.4 ± 3.90 | -71.8 ± 5.18 | -1.6 ± 0.58 |
>
> We assess stability in Llama-3.1-8B by checking whether the heads most consistently identified by greedy search (Table 9) are recovered across seeds of our compressed sensing method. For GSM8K, across four seeds, at least one of the top two heads are present in each set of ranked top 5 heads. For MBPP, the top head is found in two out of the four sets. For swearing, the top head is ranked first for all four seeds. In rhyming, the top head is present in all four sets and ranked first in three.
>
> ## Limitations and W5
> We will expand our limitations section to include the additivity assumption, which is sufficient for our setting despite ignoring interaction effects, and the opportunity for additional statistical analysis including hypothesis testing.

---

> > ### Author Rebuttal · Reviewer_GPH3 · 2026-04-03
> >
> > Thanks to the authors for the response.
> >
> > I am satisfied by the addition of CIs and the promise to include a detailed discussion of feature attribution.
> >
> > However, I am unsatisfied with several things, mostly related to the *Comparison with ProxySPEX* and *More Evals and Stability* section of the rebuttal.
> >
> > 1. It is an incorrect characterization to say "ProxySPEX aims to reconstruct the entire output signal". This is not the case at all. ProxySPEX is also extracting a sparse set of important features (this is what you call capability localization) just also accounting for interactions between heads. Reconstructing the output signal is simply a metric (of many) used for evaluation.
> > 2. For the attention head task, Fig 10 in the ProxySPEX paper, the evaluation is very similar -- heads are removed and the accuracy is assessed, expect they remove the least important heads rather than the most.
> > 3. I feel that the experiments you did with respect to ProxySPEX were somewhat lazy, though I understand this given the time constraint. This is not an on-line task, but rather a scientific experiment, so taking more samples is not a huge deal, as long as it is still feasible overall. In an ideal world, I would have liked to see if any performance is being "left on the table" by running ProxySPEX for a sufficiently large budget... with enough data it basically must converge to a "better" solution that factors in interactions. This essentially tests the linearity assumption.
> > 4. Not impressed with the stability experiments (e.g. answer to Q2). Ideally this could be done scientifically. For example, in ProxySPEX Fig. 8, the authors report Recall@k vs. number of evals used for training.
> >
> > After further reading, I feel that this work is actually quite strongly connected to ProxySPEX. For me to feel comfortable with this paper and consider raising the score to accept, I would like to see the following:
> >
> > 1. The authors acknowledge that ProxySPEX already uses compressed sensing (LASSO and the ProxySPEX algorithm) to identify sets of important heads, so they are not the first to introduce this.
> > 2. If possible use ProxySPEX, with a big enough budget, to see if the linearity assumption results in any loss.
> > 3. Proper stability experiments like ProxySPEX Fig. 8. (You can keep increasing the #evals until it is stable enough). This would significantly strengthen the claims.

---

> > > ### Author Response · Authors · 2026-04-08
> > >
> > > We would like to thank the reviewer for their follow up comments and for engaging with our work. We are glad they are satisfied with the CI results and our plan to engage further with feature attribution literature. We appreciate the deeper exploration of similarities to ProxySPEX and agree that the analysis suggested by the reviewer can strengthen the paper. We address each of the three points below and hope that it warrants a reconsideration of the score.
> > >
> > >
> > > ### Acknowledging ProxySPEX [Q1]
> > > We will add a thorough discussion of ProxySPEX, including a discussion of the similarities and differences with our method. We acknowledge that ProxySPEX utilizes a compressed sensing-based approach for feature identification, including identification of attention heads, and will ensure that our novelty claims are consistent with the prior work. In particular, our contribution is demonstrating a general and efficient first-order stratified sampling based compressed sensing approach which can effectively localize a sparse set of task-specific attention heads.
> > >
> > > ### ProxySPEX with a larger budget [Q2]
> > > We run ProxySPEX with a larger budget of 500. We use 100 samples of GSM8K, search for heads within layers 14, 15, and 16, and consider up to third order interactions. We obtain scores for all heads within the specified layers. We zero ablate the top five identified heads and evaluate the resulting model. We find that ablating these heads results in significant drops in GSM8K and Arithmetic performance and minimal drop in general language abilities. The identified heads are L15H14, L14H19, L15H13, L14H21, L15H29. We directly compare these results to those obtained by running our method. With a lower budget of 300 evals, our method matches ProxySPEX’s performance. Additionally, this lower budget was sufficient to search for heads throughout the entire model (1024 heads) rather than the limited set which ProxySPEX searched over (96 heads).
> > >
> > >
> > > | | # Evals | GSM8K | Arithmetic | Gen Avg |
> > > |---|---|---|---|---|
> > > | Baseline | - | 78.47% | 85.30% | 71.66% |
> > > | ProxySPEX | 500 | 36.54% | 24.12% | 76.36% |
> > > | ProxySPEX Δ | - | -41.93% | -61.18% | -0.30% |
> > > | Ours | 100 | 44.33% | 39.18% | 70.71% |
> > > | Ours Δ | - | -34.14% | -46.12% | -0.95% |
> > > | Ours | 300 | 37.33% | 24.09% | 70.54% |
> > > | Ours Δ | - | -41.14% | -61.21% | -1.12% |
> > >
> > >
> > >
> > > ### Stability experiments [Q3]
> > >
> > > We use the top 10 heads identified by our one-shot greedy method as the ground truth. We then compute the Recall@2, Recall@5, and Recall@10 for three seeds within each number of evals. We average the recall across the three seeds. The top 2 heads are recovered at the highest rates which is consistent with our observation that the top two heads contribute to most of the performance degradation.
> > >
> > > | # Evals | Recall@2 | Recall@5 | Recall@10 |
> > > |---------|----------|----------|-----------|
> > > | 100     | 0.500    | 0.333    | 0.200     |
> > > | 200     | 0.667    | 0.400    | 0.267     |
> > > | 300     | 0.667    | 0.533    | 0.267     |

---

### Decision · Program_Chairs · 2026-04-30

**Decision:**

Accept (regular)

**Comment:**

This paper presents an efficient compressed‑sensing approach for identifying small sets of attention heads whose ablation selectively degrades task performance, and documents this phenomenon across multiple models and benchmarks.
Reviewers generally find the method useful, well executed, and easy to adopt, and value the empirical finding that a small number of heads can be task‑critical with limited damage. The main point of disagreement concerns positioning and claims, particularly the strength of the “capability localization” framing, stability of recovered head sets under limited evaluations, and novelty relative to ProxySPEX and feature‑attribution work. The rebuttal addresses many technical concerns (confidence intervals, stability metrics, stronger ProxySPEX comparisons, and clarification of necessity vs. sufficiency), and multiple reviewers maintain strong support.

I recommend Accept. But the authors should incorporate the following: (i) clarify the localization claim throughout (capabilities are highly dependent on small head subsets, not uniquely localized or sufficient), including reconsideration of the title/abstract; (ii) more clearly position the method relative to ProxySPEX and feature attribution, with precise novelty claims; (iii) report stability/recall analyses and discuss; and (iv) release code and clarify the experimental recipe.